# A combined storyline-statistical approach for conditional extreme event attribution

Dalena León-FonFay<sup>1,2</sup>, Alexander Lemburg<sup>3</sup>, Andreas H. Fink<sup>3</sup>, Joaquim G. Pinto<sup>3</sup>, and Frauke Feser<sup>1</sup>

Correspondence: Dalena León-FonFay (dalena.leon@hereon.de)

# Abstract.

Quantifying the influence of anthropogenic global warming on extreme events requires both physical and statistical understanding. We present a framework combining two complementary conditional attribution methods: spectrally nudged storylines and flow-analogues. Applied to the 2018 Central European heatwave, storylines project an area-mean intensification of 1.7 °C per degree of global warming. Despite no detected changes in atmospheric blocking, the flow-analogue approach further indicates that heatwaves exceeding the storyline-projected intensities become far less rare at their corresponding warming levels than the factual 2018 event was under present conditions. Specifically, the 2018 heatwave, with an intensity of 2.2 °C and a return period of 1-in-277-years today, becomes a 6.6 °C event with a 1-in-26-year probability in a +4K world. We conclude that this combined framework is promising for climate change attribution of individual extreme events, offering both a physical assessment of anthropogenic warming and its associated likelihood while accounting for potential shifts in atmospheric dynamics.

## 1 Introduction

The number of extreme weather and climate events has increased in recent decades (IPCC, 2021). This is particularly the case for heatwaves, whose intensity, frequency, and length have increased in many parts of the world (IPCC, 2021; Barriopedro et al., 2023; Rousi et al., 2023; Russo et al., 2015). Heatwaves –periods of high temperature over multiple, consecutive dayspose a severe threat to ecosystems, agriculture, other economic activities, and human health (Zuo et al., 2015). In the midlatitudes, the development of heatwaves is typically associated with the occurrence of atmospheric blocking (Kautz et al., 2022). The question arises in how far these observed changes are related to human activity. Indeed, extreme event attribution to human-induced (anthropogenic) global warming (Trenberth et al., 2015; Otto, 2023) is crucial for informing and motivating policies aimed at mitigating climate change. The main idea of extreme weather attribution is to identify the role of external forcings on the change of occurrence and characteristics of the event under study (Hegerl et al., 2010). Many attribution studies clearly state that human activities are affecting the frequency and intensity of extreme events (e.g., Eyring et al., 2021; Seneviratne et al., 2021; León-FonFay et al., 2024). However, the confidence of such a statement depends on the type of event.

<sup>&</sup>lt;sup>1</sup>Institute of Coastal Systems, Helmholtz-Zentrum Hereon, Geesthacht, Germany

<sup>&</sup>lt;sup>2</sup>School of Integrated Climate and Earth System Sciences (SICSS), Universität Hamburg, Hamburg, Germany

<sup>&</sup>lt;sup>3</sup>Institute of Meteorology and Climate Research Troposphere Research (IMKTRO), Karlsruhe Institute of Technology (KIT), Karlsruhe, Germany

50

Temperature-related extremes, such as heatwaves, are strongly linked to long-term warming trends and offer higher attribution confidence due to the clear human footprint (National Academies of Sciences, Engineering, and Medicine, 2016). In contrast, non-heatwave events shaped by complex dynamical processes -in particular when associated with modification of the large-scale atmospheric circulation like windstorms- imply greater challenges for attribution, due to higher internal variability and a lower signal-to-noise ratio (Shepherd, 2014).

Shepherd (2016) divides attribution methods into 'storyline-based' and 'risk-based' approaches. The former consists of examining a plausible physical unfolding of a specific event of study in an alternative thermodynamic background (either a counterfactual or future warmer worlds) compared to the observed one. The latter consists of identifying the change in the probability of occurrence of an event of a certain magnitude in samples representing a factual (observed) world, a counterfactual (no global warming), or future warmer conditions. From the storyline approach, one usually addresses questions like, "How would this specific event have changed in the absence of global warming?", while the risk-based approach aims to reply: "How likely is this type of event now compared to a counterfactual past or possible futures?". Independently, both approaches provide different answers to the role of anthropogenic global warming on future events, but they are rarely combined to bring a more comprehensive view of individual extreme events (Shepherd, 2016). Other studies have aimed to merge these approaches by deriving storylines from flow-analogue reconstruction and parallelly applying unconditional risk-based attribution (World Weather Attribution-style) (e.g., Qian et al., 2023; Ye et al., 2025), considering classes of similar extreme events, but not a specific historic extreme event. In this article, we present a novel combined attribution approach conditioned on the atmospheric circulation of a specific, high-impact extreme event, focusing on the thermodynamical aspect of anthropogenic global warming. For the storyline approach, we use spectrally nudged storylines (van Garderen et al., 2020), which are highly conditioned to the specific large-scale pattern that shapes an extreme event. The traditional risk-based approach, like the World Weather Attribution method (Philip et al., 2020), would not align with our purpose of focusing on a specific event due to its unconditioned nature. Instead, a conditional statistical approach like the flow-analogue method (e.g., Zorita and Von Storch, 1999; Yiou et al., 2017) allows to identify a subset of events under similar circulation patterns such that the dynamical uncertainty is also reduced, focusing on the thermodynamic aspects. In that way, one can extend the key research question to: To what extent has human influence altered both the magnitude and the likelihood of extreme events beyond natural variability, and how might these events evolve under future warming scenarios? (Trenberth et al., 2015). This framework represents a first step towards a conditional physical-statistical extreme event attribution.

The Spectrally Nudged Storylines approach (van Garderen et al., 2020) emerged as a method that isolates the thermodynamic influence of global warming on specific extreme events, while minimizing uncertainties related to dynamical variability (Shepherd, 2014; Trenberth et al., 2015; Feser and Shepherd, 2025). This methodology simulates different storylines of the same event by constraining large-scale atmospheric circulation to resemble observed dynamical conditions (spectral nudging) (von Storch et al., 2000), under the assumption that such circulation patterns could also occur in different climates. The thermodynamic conditions are then modified to represent the event under counterfactual, factual, and future global warming levels (Shepherd et al., 2018; van Garderen et al., 2020). This allows the role of anthropogenic global warming to be attributed independently of dynamical variability, which is often a source of uncertainty in climate model simulations (Shepherd, 2014). For

65

instance, case studies following this methodology suggest that the observed heatwave of 2022 led to a 5.7 °C exceedance of the climatological temperature over the Balkans, while in a +2K world, one could expect Poland to experience up to 7.6 °C above the climatology (Feser et al., 2024). For the July 2019 heatwave over Europe, studies suggest warming rates ranged between a factor 2–3 for Central Europe, resulting in up to 12 °C warming for a world with +4 °C global average temperature (+4K climate) (Sánchez-Benítez et al., 2022; Klimiuk et al., 2025). While the method performs successfully in evaluating individual study cases under different plausible scenarios, the question remains of how likely such projections are.

The flow-analogue method (Zorita and Von Storch, 1999; Vautard et al., 2016; Yiou et al., 2017) seeks to disentangle the dynamics from thermodynamics and attribute their contribution to an extreme event, but using a probabilistic approach. Instead of isolating a specific event, it identifies similar large-scale circulation patterns to the one of the extreme event to be analyzed across a large ensemble of simulations, assessing thus how such a class of events changes in present and alternative climates (e.g., Jézéquel et al., 2018; Vautard et al., 2016). For instance, Yiou et al. (2008) applied the flow-analogue method to study extreme temperature and precipitation events in Europe, demonstrating how changes in weather regimes could influence the occurrence of such extremes under climate change scenarios. Similarly, this approach has been used to examine the dynamics of European heatwaves, highlighting shifts in circulation patterns that contribute to their increased frequency (Jézéquel et al., 2018). While this method is less suited to attributing the precise influence of anthropogenic warming on a single event, it provides robust estimates of the likelihood of similar events occurring under varying climate conditions, important for risk assessments (Trenberth et al., 2015).

When combined, these two methods provide complementary insights into extreme event attribution and projection. The storyline approach offers a physically detailed narrative of how a specific event evolves under alternative levels of global warming, but is limited in its ability to assess probabilities due to the small number of simulated events (Feser and Shepherd, 2025). Conversely, the flow-analogue method, based on large ensembles, offers probabilistic estimates of risk while lacking the event-specific physical detail of the storyline approach. By conditioning both methods on the observed circulation pattern, this framework enhances causal inference by isolating the thermodynamic influence of anthropogenic warming while quantifying changes in the likelihood of dynamically comparable events. Together, they enable both a physical understanding of event intensification and a probabilistic assessment of how its likelihood changes with global warming for a given dynamical system.

We illustrate this framework using the July 2018 Central European heatwave, one of the strongest European heatwaves on record in terms of magnitude, spatial extent, and legacy effects (Rousi et al., 2023; Knutzen et al., 2025; Xoplaki et al., 2025). The event was marked by persistent atmospheric blocking over Scandinavia and Central Europe, exceptionally high sea surface temperatures, and low soil moisture in Spring, which contributed to sustained heat and drought conditions across large parts of Europe (Lhotka and Kyselỳ, 2022; Yiou et al., 2020; Rousi et al., 2023; Knutzen et al., 2025). The heatwave was exceptional in the sense of its long duration, prolonged drought, and prevalence (Rousi et al., 2023; Sagen, 2020). In Germany, summer temperatures set meteorological records as it was the hottest one over the Northern and Eastern part and the driest one in the middle of Germany at the time (Imbery et al., 2018; Rousi et al., 2023). Overall, the 2018 summer was very hot and dry, more than 2.5 °C warmer than average in many regions, leading to severe agricultural and ecological impacts in the following years, including widespread drought-induced tree mortality in Central Europe (Schuldt et al., 2020; Knutzen et al., 2025).

100

In this study, we apply both the storyline and flow-analogue methods to this event to evaluate the potential of a combined attribution methodology for extreme events. A schematic of the framework can be seen in Fig. 1. Using spectrally nudged storylines, we quantify the thermodynamic response of the 2018 heatwave to anthropogenic global warming under alternative climate conditions. In parallel, we apply the flow-analogue method to the MPI-ESM-LR Grand Ensemble (50 members) (Olon-scheck et al., 2023) to evaluate changes in the frequency of the associated circulation pattern and the occurrence of heatwaves with similar characteristics across different levels of global warming. Finally, we join these methods to estimate the likelihood of projected heatwaves in the storyline approach in future warming levels. The hypothesis is that this framework not only complements the individual strengths of each method but also enhances the robustness of the storyline approach by complementing its physical narrative with a probabilistic perspective and quantifying the uncertainty in event occurrence. A description of each dataset and method used can be found in Sect. 2 and 3. In the Results Sect. 4, we show an application of the methodology to the attribution of the influence of global warming on heatwave intensity.

Figure 1. Framework schematic. Given an extreme event of interest, the spectrally nudged storyline (1) and flow-analogue (2) approaches are applied in parallel. The storyline approach generates physically consistent versions of the event to assess changes in its magnitude due to global warming. In this case, the stars represent the evolving mean intensity of the event across storylines. The flow-analogue method constructs probability distributions of heatwave mean intensity associated with circulation analogues, capturing both changes in the frequency of the flow pattern and the likelihood of similar events occurring at future warming levels. The combined application (3) involves taking the projected magnitudes from the storyline approach and evaluating their probability of occurrence using the distributions derived from the flow-analogue method. This allows for an assessment of how likely the projected event magnitudes are under their corresponding level of global warming.

#### 105 2 Data

130

#### 2.1 ERA5

As a reference dataset, we use ERA5 (Hersbach et al., 2020), the latest generation of a global reanalysis product provided by the European Centre for Medium-Range Weather Forecasts (ECMWF). This dataset provides a gridded best possible estimate of the state of the atmosphere by combining short forecasts from ECMWF's Integrated Forecasting System (IFS) Cycle 41r2 and the so-called 4D-Var data assimilation scheme fed by numerous observations. For characterizing the observed 2018 heatwave in terms of near-surface temperature as well as the large-scale atmospheric flow, we used 2m maximum temperature as well as 500hPa geopotential at a daily temporal resolution on a spatial grid of 0.25° x 0.25° from the ERA5 post-processed daily statistics on single levels dataset (Hersbach et al., 2023).

# 2.2 Global spectrally nudged storylines

- In this study, we first use a storyline approach to estimate the impact of climate change on the July 2018 event. The dataset consists of 5 storylines, namely:
  - Counterfactual: a world without the influence of anthropogenic global warming, corresponding to pre-industrial times (1850–1920).
  - Factual: the world under the present level of global warming (2015–2025).
- 120 +2K, +3K, +4K: the world under +X K degrees of global warming with respect to pre-industrial times (1850–1920).

Each storyline represents a physically consistent, plausible scenario in which our observed climate could have developed under an alternative thermodynamic background, simulated as described in detail in van Garderen et al. (2020). The storylines were simulated using the atmospheric general circulation model ECHAM6.0 -atmospheric component of the MPI-ESM model-(Stevens et al., 2013; Giorgetta et al., 2013) (T255, 95 levels), which integrates the land vegetation model JSBACH for land processes. The spectral nudging of vorticity and divergence towards NCEP-NCAR reanalysis (Kalnay et al., 1996; Kistler et al., 2001) ensures that the large-scale weather pattern (nudged for wave numbers n<38, >1000km) stays close to the observed one in all storylines. The storylines then mostly differ in their thermodynamics, diminishing the influence of internal variability. To impose different levels of global warming for each storyline, sea surface temperatures and greenhouse gases are prescribed according to the desired level of global warming (since these variables are influenced by anthropogenic forcing with a high level of certainty (Eyring et al., 2021; van Garderen et al., 2020)). As a result, all 5 storylines listed above are available for the same period, running from 2015 to the present day under different warming conditions. Each storyline has 5 members, such that the spin-up simulations for each member per storyline were started at different dates, in consecutive weeks, to account for model and initial condition uncertainty.

Hence, the chosen model setup nudges the model to reproduce the observed evolution of the synoptic-scale flow (e.g, atmospheric ridges, troughs, and blockings), being able to react to imposed thermodynamic changes and allowing changes to realistic local weather events.

#### 2.3 MPI-ESM-LR

For the statistical analysis, we use transient simulations from the MPI-ESM-LR grand ensemble (50 members) (Olonscheck et al., 2023) climate model under the SSP5-8.5 emission pathway and historical run. The grand ensemble is simulated using the Max Planck Institute Earth System Model version 1.2 (MPI-ESM1.2), in the low resolution (LR) setup (T63, 1.8° atmosphere; GR15, 1.5° ocean) (Mauritsen et al., 2019). Due to the high-frequency temporal output (3 to 6 hourly) availability for its 50 realizations, the model provides a sample size large enough to work with flow-analogues and extreme events.

The dataset for each degree of global warming is defined as the 20 years centered around the year where the 20-year running average of global mean temperature surpasses the desired temperature anomaly above pre-industrial times (1850–1920). For example, if +2K is reached in 2033, the period 2023–2043 is taken. This is done for each ensemble member, such that, in total, each global warming level has 1000 years (20 years x 50 members) of data.

#### 3 Methods

#### 3.1 Attribution using Spectrally Nudged Storylines

As described Sect.2.2, the spectrally nudged storylines reproduce a specific event of interest under different warming levels, in our case the July 2018 heatwave. Since the only differences between them are given by the background thermodynamic conditions induced by anthropogenic global warming, all differences between them can be directly attributed to human-induced global warming. More specifically, all changes between the present (factual) or future (+2K, +3K, +4K) storylines with respect to the counterfactual storyline address: *What is the influence of anthropogenic global warming on the extreme event of study?* This approach thus exploits communication of the consequences of global warming through historical events rooted in collective memory (Feser and Shepherd, 2025).

# 3.2 Flow-analogues

To characterize the large-scale circulation pattern associated with the 2018 heatwave over Central Europe (July 24th – August 10th), we use ERA5 geopotential height (GPH) anomalies at 500 hPa (with respect to the 1985–2014 mean climatology). A 5-day sequence (July 31st–August 4th) preceding the heatwave peak is extracted from ERA5 as the reference pattern (See Supplementary Fig. S1–S2). Analogueous circulation patterns are identified using detrended GPH anomalies from the MPI-ESM-LR Grand Ensemble, restricted to summer months (June–August). Following Vautard et al. (2016) recommendations to take an appropriate region for the flow-analogues, the reference flow pattern used is taken over an extended region (20° W, 20° E, 40° N, 65° N) surrounding the region of the event of interest (Central Europe: 3° E, 18° E, 44° N, 55° N). The region is large

165

enough to capture either the synoptic-scale ridge and trough pattern or blocking anticyclones, which are mainly responsible for heatwave formation. On the other hand, it is not too large to ensure that the analogues are representative of Central European heatwave conditions. We further decided not to center the box around the Central European focus region, but to shift it by some degrees westward, as the important synoptic-scale features and high gradients in the geopotential field tend to be located further upstream.

Analogue candidates are selected based on the smallest Euclidean distance calculated from every possible 5-day sequence within the MPI-ESM data that matches the reference 5-day flow pattern. To avoid using shifted time windows of the same flow-analogues, the N closest matches are taken with a minimum 15-day separation between events. From the N closest matches, only those with a spatial correlation greater than 0.8 compared to the time-averaged reference pattern are retained (See Supplementary Fig. S3 for correlation matrices between analogues and ERA5).

The analogue selection process is applied to subsets of the MPI-ESM-LR Grand Ensemble, grouped by global warming level (counterfactual, factual, +2K, +3K, +4K), where each subset consists of 20-year time slices per ensemble member. With 50 ensemble members, a total of 1000 years of data is available per warming level, providing a large and robust sample for analogue selection. An analysis of the analogue detection performance and quality is provided in Supplementary Figs. S3–S5.

# 3.3 Heatwave definition

We define a heatwave as an event where the daily maximum temperature (TX) exceeds the climatology's 95th percentile 180 (TX95) of the given calendar day for more than 3 consecutive days. The 95th percentile threshold is computed for each calendar day using a 15-day centered moving window over the 1985–2014 climatological period. This definition holds for both field-averaged and local analysis.

The heatwave intensity is defined as the temperature anomaly exceeding the 95th percentile of the climatology during the heatwave days (n)(Barriopedro et al., 2023).

$$\text{HW intensity} = TX_i - TX95_i \qquad \qquad \text{HW mean intensity} = \sum_{i=1}^n \frac{TX_i - TX95_i}{n}$$

# 185 3.4 Return periods

To attribute the likelihood of the event occurring under present and alternative warming levels, we calculate the return periods of the July 2018 heatwave using subsets of analogue events identified under different warming levels with the flow-analogue approach (Sect. 3.2). The return period represents the inverse probability of an event surpassing a specific magnitude (return level). In our study, we are interested in the probability of occurrence of an event of a certain magnitude given a circulation pattern of interest (flow-analogues). This recurs to Bayesian probability, such that let P(D) be the probability of the flow-analogue to occur, and P(E|D) the probability of the event of interest to occur within such a circulation pattern. The occurrence probability of the event of interest would then be P(E,D) = P(D)P(E|D). Finally, the probability of an event of magnitude M, under the circulation pattern of interest D, in a given global warming level (GWL) will be defined as  $P_{M,GWL} = P(E,D)_{GWL}$ .

To better represent return values within a 95% confidence interval, the distribution functions of the subset of temperatures related to the flow-analogues are fitted using a generalized extreme value distribution and performing 1000 samples with bootstrap-resampling.

# 4 Results

# 4.1 Storyline attribution of the influence of anthropogenic global warming on the 2018 heatwave.

Figure 2. The 2018 Central European heatwave. a) Mean geopotential height anomaly (gph@500hPa) and streamlines based on 500 hPa mean wind components during the event of interest (July 24th–August 10th). The green box encloses the region (20° W, 20° E, 40° N, 65° N) used for flow-analogue detection. b) Maximum near-surface temperature anomaly with respect to mean climatology during the heatwave event. The blue box encloses the region of interest (Central Europe: 3° E, 18° E, 44° N, 55° N). c) Daily maximum temperature time series spatially averaged over Central Europe for each storyline (CF: counterfactual, factual, +2K, +3K, +4K), counting on 5 members each (thin lines). The dashed line corresponds to the 95th percentile of the 1985–2014 climatology. The black solid line corresponds to ERA5 data for comparison to the factual world. The yellow shaded region encloses the length of the event in the factual world (July 24th–August 10th).








We analyse the features of the 2018 heatwave over Central Europe (blue box in Fig.2(b): [3° E, 18° E, 44° N, 55° N]) focusing on the core period of July 24th to August 10th (Fig. 2). The flow-pattern for the entire event can be found in Supplementary Fig. S6. In agreement with the literature, this event is characterized by persistent atmospheric blocking, strong geopotential height (GPH) anomalies, and thus an enduring disruption of the westerly zonal flow over Scandinavia and Northern Germany (Lhotka and Kyselỳ, 2022; Rousi et al., 2023) (Fig. 2(a)). These are the regions where the heatwave peaked in July, leading to high temperature anomalies compared to mean climatological values (Fig. 2(b)). Averaged over the heatwave duration, these anomalies reached up to 10 °C in Germany and up to 8 °C in some regions in Scandinavia. The time series in Fig. 2(c) corresponds to the field-averaged maximum near-surface temperatures over Central Europe (blue box) per storyline, which shows an increase in temperature and duration of the heatwave with global warming. The corresponding ERA5 temperatures for the region (black solid line) coincide with the factual storyline, showing an accurate representation of the observed event. The climatological period (1985–2014) was also simulated with spectral nudging as in the factual storyline for consistency. We limit most of our analysis to this region for two main reasons: the first one being to perform a regional study focused on Germany due to its large local impacts, and the other reason is methodological rather than impact-based; the use of a smaller region allows for a better chance to find close analogues of high quality over an extended region (green box in Fig.2(a)) that captures the large-scale circulation pattern behind the heatwave event (Jézéquel et al., 2018).

Even though the heatwave was particularly extreme over Central Europe and Scandinavia, it also affected other regions in Europe with less intensity. In Fig. 3(b), it can be seen how in a factual world, the highest heatwave intensities occurred in Northern Germany. Here we define intensity as the exceedance (in °C) of the local 95th percentile of the climatological daily maximum temperature (see Sect. 3.3). Local maximum intensities of around +9 °C (Fig. 3(b)), and a mean intensity of 2.2 °C (Fig. 3(f)) were reached over the region of interest during the days of the observed (factual) event. In the absence of anthropogenic global warming, the counterfactual storyline also shows the presence of a heatwave over a less extended region (Fig. 3(a)) and period of time (Fig. 3(f)). This counterfactual event would have also affected Northern Germany and Scandinavia, but the heatwave's mean intensity would be limited to 0.5 °C (Fig. 3(f)), and local maxima would have reached at most a 5 °C intensity (Fig. 3(a)). Hence, human activity amplified the observed heatwave's characteristics, but the heatwave would have still developed under pre-industrial conditions. As global mean temperature increases, the heatwave mean intensity increases from 0.5 °C in a counterfactual world to 6.6 °C under a +4K level of global warming (Fig. 3(a–e)). The heatwave also extends from affecting mainly Northern Europe, to affecting the whole European region, starting from a +2K level.

The main contribution of anthropogenic global warming to temperature increase over the region of interest can be evidenced in Fig. 4 using warming rates (t2m increase per degree of global warming). The 5-day running mean warming rate for maximum, mean, and minimum temperature increases towards the heatwave event, reaching values of ~+1.7 °C per °C of global warming at the center of the heatwave event for maximum near-surface temperature (Fig. 4(a)). The mean maximum temperature of the event (Fig. 4(b)) in a counterfactual world would be 26.7 °C, while the observed heatwave (factual world) had an average maximum temperature of 28.8 °C. This indicates a +2.1 °C increase in maximum temperature in the observed heatwave compared to the counterfactual, which can be directly attributed to anthropogenic global warming. Temperatures rise to 33.3 °C in a +4K world, under a warming rate of 1.66 °C per °C of global warming. The local warming rate in maximum




**Figure 3. Heatwave intensity**. a—e) Local maximum heatwave intensity during the heatwave event for each storyline. f) Time series of daily heatwave intensity over the region of interest (Central Europe) enclosed in blue boxes. The stars in the time series plot are the heatwave mean intensity per storyline (magnitudes shown in the upper left corner of the maps).

temperature (Fig. 4(c)) corresponds to the local trend per grid cell of the mean maximum temperature during the heatwave event for each storyline. Even if Northern Germany was the most affected region by the heatwave, Central and Southeastern Germany show the largest local warming rate for maximum temperatures, scaling by 2 °C per °C of global warming (Fig. 4(b)). In general, we find an amplified warming rate response to increased global mean temperature, meaning that the local trends are at least 1 °C per °C of global warming and more (blueish colors), with most of the region affected by a warming rate of 1.5 °C per °C of global warming or more (orange-redish colors). A similar behavior is seen for the 5-day running warming rate during summer months (Fig. 4(a)), an overall amplified response is seen for minimum, mean, and maximum near surface temperature for most time windows.

The storyline method enables specific, strong arguments about the absolute contribution of the thermodynamic influence of anthropogenic global warming to changing magnitudes of an extreme event's characteristics. Here, we state that the 2018 heatwave with a mean intensity of 2.2 °C would have been 6 days shorter and with a 0.5 °C mean intensity in the absence of human-induced global warming (Fig. 3(a)). Regarding future warming scenarios, an event developing under the same atmo-

Figure 4. Warming rate: 2 meter temperature increase per degree of global warming. a) 5-day running average warming rate for minimum (blue), mean (green), and maximum (red) near-surface temperature spatially averaged over Central Europe. Color shading corresponds to the range covered by the 5 available members. b) Mean warming rate during the July 2018 heatwave event per degree of global warming (counterfactual, factual, +2K, +3K, +4K) spatially-averaged over Central Europe. c) Local warming rate of the event's mean maximum temperature over the region of interest (Central Europe).




spheric conditions would result in a heatwave 1.9 °C more intense than in present times in a +2K world, up to 4.4 °C more intense in a +4K world, where average maximum 2m-temperatures could reach 33.3 °C for the heatwave event. However, as compelling as these event-based narratives are, it is important to recognize that every heatwave is unique. Here, the flow-analogue method becomes an essential complement to the storyline approach. While we may never witness the exact same event twice, we can search for analogue events with similar atmospheric conditions under future warming levels. In this way, the analogue approach aims to bridge the gap between storyline-driven projections and real-world probabilities.

## 4.2 Dynamical comparability between approaches

Studies argue that it is challenging to find good analogues when the event is too intense (Qian et al., 2023), or due to changing dynamics in the future (Thompson et al., 2024; Vautard et al., 2023). This limitation of the analogue approach is particularly evident when used in the conventional way (e.g., Wang et al., 2023; Yiou et al., 2017), where the analogues aim to reproduce not only the flow pattern of interest but also the pattern of the variable of interest to reconstruct the observed event. In our case, we are interested in the resemblance of the large-scale flow only, featuring temperature fields that do not necessarily reproduce the observed one. We deliberately let the associated variable vary to be able to construct a distribution function out of all the possible temperatures fields related to the given circulation pattern.

In Fig. 5, we illustrate the similarity of the circulation pattern of interest across the storyline and flow-analogue approaches, relative to the observed pattern (based on ERA5). Both the simulated pattern in the storyline approach under present conditions and the mean of all analogues identified during the equivalent factual period across the 50 members of the MPI-ESM-LR GE using the analogue selection process described in Sect. 3.2 accurately represent the flow pattern of interest. Given the comparable dynamics in both approaches, we can attribute changes in the July 2018 heatwave more confidently to the thermodynamic component of anthropogenic global warming. In supplementary Figs. S3 and S4, we further evaluate the quality of the identified analogues and their mean behavior at different levels of global warming, relative to the ERA5 reference circulation. These results demonstrate the recurrence and robustness of the analogues, as well as their similarity to the reference flow, regardless of the warming level.


Figure 5. Comparison of observed (ERA5), storyline (factual), and analogue-derived (factual period MPI-ESM-LR GE) circulation pattern. All three plots are based on the 5-day mean (July 31st–Aug. 4th) GPH anomaly at 500hPa with respect to the mean climatology 1985–2014.

# 4.3 Likelihood of analogue events in present and warmer worlds.

Figure 6. Temperature anomaly distribution for best flow-analogues of the 2018 HW. Each distribution curve corresponds to a different level of global warming, accordingly to the nomenclature in the storyline method (counterfactual, factual, +2K, +3K, +4K). The values portrayed correspond to the 5-day mean temperature anomaly (relative to the 95th percentile of the 1985–2014 climatology) for each analogue identified. The sample size N (see label) refers to the number of best analogues (closest Euclidean distance and mean spatial correlation>0.8) found in these global warming levels out of the 1000 years dataset from the MPI-ESM-LR GE. The stars show the mean intensity of the heatwaves projected by the storyline method (as in Fig. 3).

Based on the flow-analogue selection process described in Sect. 3.2, we built subsets of N best analogues for each level of global warming comparable to the ones in the storyline approach. These analogues represent events which feature a similar








large-scale flow evolution during the 5 days preceding the heatwave peak (July 31st–August 4th ) (See Supplementary Figs. S1–S2). In Fig. 6, the sample size N corresponds to the number of best analogues identified from the initial 1000 years (50 members x 20 years) dataset for each global warming level in the MPI-ESM-LR GE (see Sect. 2.3). Roughly N $\approx$ 250 flow-analogues are consistently identified across all warming levels (Fig. 6: see label). This suggests that the circulation pattern linked to the 2018 heatwave remains roughly equally likely to occur in a warmer climate, regardless of increases in global mean surface temperature, in agreement with previously documented literature (e.g., Yiou et al., 2020; Davini and d'Andrea, 2020). A common critique of the storyline method is its assumption that the same large-scale atmospheric circulation patterns will occur in future climate scenarios, even though studies project dynamical changes in the atmosphere (Vautard et al., 2023), which could challenge the validity of such assumptions. However, our results provide evidence that, in this specific case study, the circulation pattern in question could be expected to occur with a  $\sim$ 1-in-4 year probability (P(D)) regardless of the change in global mean surface temperature. In cases where the circulation pattern of a given case study can no longer be identified in future scenarios, this should not be regarded as a drawback of the method. Rather, it provides valuable information, allowing us to reject the possibility of a warmer storyline of the event occurring under future conditions.

Figure 6 shows the distribution of 5-day mean temperature anomalies of the analogues, defined as the field averaged daily maximum temperatures relative to the climatological 95th percentile  $(\text{mean}(TX_i - TX95_i))$  for Central Europe. We refer to these values as t2m anomalies rather than heatwave intensities (although they share the same definition; see Sect. 3.3), since not all identified flow analogues develop into a heatwave. In Fig. 6, the black line defines the limit between analogues that evolve into a heatwave (anomalies > 0), and those that do not (anomalies 




probability of the atmospheric circulation pattern to occur P(D), resulting in  $P_{M,GWL}$  (see Sect. 3.4), presented in Fig. 7 in terms of return periods.

**Figure 7. Heatwave intensity's return periods.** a) Return period of the factual 2018 HW per degree of global warming. b) Return periods of heatwave intensity for each global warming level (counterfactual, factual, +2K, +3K, +4K). Solid lines show return levels obtained through GEV fitting of 250 analogue events scaled by their 1-in-4 years occurrence out of the 1000 years sampled (50 members x 20 years). Shaded regions enclose the 95% confidence interval (lower bound: 2.5th percentile, upper bound: 97.5th percentile) obtained by a 1000-sample bootstrap resampling. Stars show the return period equivalent to the projected intensity by the storyline approach.

The key question in statistical attribution methods is to assess the change in the likelihood of the observed event, in this case, the 2018 heatwave, occurring in the present climate compared to counterfactual or future warming scenarios. We denote the probability of the factual heatwave of magnitude 2.2 °C to occur in a given global warming level as  $P_{2.2^{\circ}C,GWL}$ . In Fig. 7, the blue star markers provide the answer to such a question; the counterfactual world has been omitted in Fig 7(a) since a heatwave of such magnitude was not identified (see Fig. 6). This is also evidenced in Fig. 7(b), where the counterfactual dataset does not reach the observed magnitude, meaning that it was not possible to occur in the absence of human-induced warming. In a factual world, the 2018 heatwave conditioned to such an atmospheric circulation pattern had a 1-in-277-years chance to occur, being a rare event in present times, with only 1.6% of the analogues surpassing the experienced mean heatwave intensity of 2.2 °C (Fig. 6). Under continued global warming, the return period of a heatwave with the same intensity would decrease exponentially (Fig. 7(a)), eventually becoming a common 1-in-5-year event in a +4K world.

The added value of this combined approach lies in its ability to quantify the likelihood that an alternative, physically consistent storyline of the heatwave would emerge under its corresponding level of global warming (Fig. 7(b)). Based on the storyline approach, the 2018 heatwave would have a mean intensity of 4.1 °C in a +2K world, with a 1-in-112 years probability



to occur  $(P_{4.1}\circ_{C,+2K})$ . In a +3K world, the mean intensity would increase to 5.4 °C, with a return period of 1-in-58 years  $(P_{5.4}\circ_{C,+3K})$ . Finally, in a +4K world, the heatwave's mean intensity would reach 6.6 °C, with a return period of just 1-in-26 years  $(P_{6.6}\circ_{C,+4K})$ . Since we are using a fixed climatology as a baseline to define present and future heatwaves, it is no surprise that the 2018 heatwave, as experienced in the present, undergoes an exponential increase in likelihood until it becomes a common event in a +4K world. However, one might initially assume that the warmer storylines simulated for future climates would remain as unlikely in their respective levels of global warming as the original heatwave was in the present. Instead, the results indicate that these projected events also become more common at their corresponding global warming levels, following an exponential trend. These return periods already take into account the probability of the atmospheric circulation pattern occurring under future climate conditions, whose frequency stays roughly constant with increasing global mean surface temperature. This indicates that due to dynamical conditions, future heatwaves have the same chance to occur. Despite that, anthropogenic global warming seems to intensify rare heatwave events in warming scenarios at a higher rate than expected and projected by the storyline approach. Even though these results may be model-dependent, it is worth emphasizing that both approaches are physically consistent, as they rely on the same model. The storylines are simulated with ECHAM, the atmospheric component of the MPI-ESM model, while the flow-analogues are extracted from scenario simulations of MPI-ESM. Including additional models for comparison in future work could provide more detailed insights on this matter.

# 5 Conclusions and discussions

Our study brings together physical and statistical narratives in conditional extreme event attribution. We used the July 2018 heatwave over Central Europe to contextualize the proposed storyline-statistical attribution approach. In this study, the storyline approach was conducted using a global spectrally nudged storyline dataset resembling the event of interest unfolding in a counterfactual, present, and future warming scenarios. The statistical approach was performed through the flow-analogues method, which conditions the analysis to the circulation pattern related to the event of study, applied to the MPI-ESM-LR Grand Ensemble (50 members) for equivalent counterfactual, present and future warming scenarios. Our results show that:

- There is an amplified mean warming rate during summer days, which intensifies during the heatwave event, reaching a
  rate of increase in maximum temperature of 1.7 °C per degree of global warming.
- Locally, an overall amplified warming rate is expected in Central Europe, with Central and Southeast Germany experiencing a warming rate surpassing the 2 °C increase per degree of increase in global warming.
  - No dynamical trends were identified for the atmospheric circulation pattern associated with the 2018 heatwave. The blocking system remains roughly equally likely to occur in present and future levels of global warming, with a probability of approximately 1-in-4 years.
- Our results suggest that the atmospheric blocking system enabling the 2018 heatwave changes from being a merely necessary factor under current climate conditions to becoming an increasingly sufficient condition (e.g., 97,1% probability in a +4K world) for the occurrence of a heatwave.



- The factual Central European heatwave of July 2018 becomes exponentially more common with global warming. The observed heatwave goes from being a 1-in-277 years event in the present, increasing in frequency until becoming a 1-in-5 years event in a +4K degree world.
- Warmer storylines of the 2018 heatwave are not as rare at their corresponding level of global warming, as the 2018 heatwave was in the present. While the heatwave intensities increase linearly per degree of global warming, their frequency exponentially increases. Specifically, the observed 2018 heatwave had a 2.2 °C intensity with a return period of 1-in-277 years in the present time, intensifying to 6.6 °C with a return period of 1-in-26 years in a +4K world.

Some former studies attributed the anomalous intensity and high persistence of the 2018 European heatwave to anthropogenic climate change. Using Earth system models, Vogel et al. (2019) concluded that concurrent hot extremes of the Northern Hemisphere in 2018 would not have been possible without human impact. In another study, Yiou et al. (2020) performed unconditional and conditional attribution to the 2018 heatwave, concluding that its likelihood and intensity increased due to anthropogenic global warming. Using a probabilistic attribution approach, Rousi et al. (2023) provided evidence that such events have already become more likely in recent decades, and are expected to occur virtually every single year in a +2K warmer world. Moreover, Wehrli et al. (2020) used storylines to attribute the 2018 heatwave and found that under similar atmospheric weather patterns, the heatwave would reach potentially health-impacting temperatures of more than 40 °C for future climate conditions of +2K.

Regarding the warming rate behavior, our results agree well with an earlier storyline-based study by Klimiuk et al. (2025). Using a different storyline set-up, they report similarly high amplified warming rates for the late July 2019 heatwave in a similar region. In their study, they conclude that summer months show an overall amplification in warming rates, with further intensification during heatwave events, reaching up to 1.9 °C per degree of global warming. Locally, warming rates for maximum temperature exhibit a comparable behavior, with most land areas in the region of interest experiencing at least a 1 °C increase per °C of global warming. These findings highlight that, even within the highly conditional nature of the storyline approach applied to individual case studies, Central Europe exhibits consistent thermodynamic responses across different extreme heatwave events.

About the recurrence of the 2018 heatwave circulation pattern in the present and future levels of global warming, our results agree with Yiou et al. (2020), who identified no dynamical trends for good analogues of the 2018 heatwave. Using the analogue method for an extreme precipitation event in July 2021, Thompson et al. (2024) found that the analogue catalog of comparable Central European cut-off lows shows a systematic eastward shift in future projections. However, our 2018 heatwave analogues do not show any systematic shifts in the positioning of the atmospheric blockings (see Supplementary Fig. S5). We only detect a general tendency for overall slightly weaker GPH anomalies in all future warming scenarios, which should not have major implications for the overall large-scale atmospheric flow. Hence, we are confident that the statistics of the respective analogue samples are not affected by systematic changes in the large-scale flow.

Concerning the increased probability of occurrence of the observed 2018 heatwave in future climate conditions, other researchers also report an increased likelihood due to global warming. In a study based on cumulative heat (defined as the







integrated temperature exceedance above a threshold), Rousi et al. (2023) estimated a 96% probability of the 2018 heatwave being exceeded in a +2K world. In another study, Felsche et al. (2024) suggest a four-fold increase in exceedance probability of the 2018 heatwave based on a multivariate analysis, becoming a 1-in-10-years event in a +2K world. These results differ in magnitude from ours, given that their results correspond to the probability of any heatwave to emerge in this region with similar characteristics -ex: cumulative heat, intensity, or multivariate analysis-, regardless of the dynamical conditions. On the other hand, our analysis quantifies the probability of having a heatwave like the one experienced in 2018, given the occurrence of an atmospheric blocking over Scandinavia (which remains very likely in future warmer worlds). These results highlight the dependence of attribution methodologies on the event's definition and remark on the different messages provided by attribution methodologies. Overall, the key message is shared: extreme events like the 2018 heatwave will eventually become normal, starting from a +2K global mean temperature anomaly.

The major key finding of this research is the exponential increase in exceedance probability of the warmer storylines of the 2018 heatwave at their own level of global warming. This result can only be achieved by combining projected storylines of the event with analogue events selected at corresponding global warming levels from large ensemble datasets. Our results suggest that heatwave intensities in the storyline-based projections scale at a slower rate than expected in the MPI-ESM-LR Grand Ensemble. For example, while the factual 2018 heatwave has an estimated return period of 1-in-277 years in the present climate, the +4K storyline version of the event corresponds to a 1-in-26 year event under +4K conditions, indicating a tenfold increase in probability. According to Feser et al. (2024), the spectrally nudged storyline approach is conservative as it represents the influence of anthropogenic global warming solely through information stored in sea surface temperature and greenhouse gases, while other variables which are likely also influenced by human activity -like aerosols- are not taken into account, representing climate change more cautiously. This may explain the higher rate of increase in the global warming signal in the MPI-ESM GE dataset compared to the simulated storylines, and also suggests that future heatwaves could be even more intense than what the storyline projections indicate.

In summary, the storyline method, on its own, provides a quantification of changes in heatwaves' characteristics due to global warming. For instance, stating how much warmer the event would be if it were to develop under warmer background conditions. These statements help in terms of communication, where the public can relate to past experiences and connect them to the severity of future scenarios under the concept: "The 2018 heatwave was already extreme, imagine it in a warmer world". On the other hand, the flow-analogue method alone provides a distribution of temperatures associated with the circulation pattern of interest and assesses how the likelihood of heatwaves similar to the observed 2018 event changes with global warming. We have demonstrated that a more complete story arises when both methods are combined. For the 2018 heatwave over Central Europe, we provided evidence that the large-scale atmospheric circulation pattern related to the extreme event does not change with increasing global warming, so that it is as likely to occur as in present times. Naturally, it is not only the atmospheric circulation pattern and global warming that trigger the occurrence of a heatwave. For such reason, not all flow-analogues evolve into one; many other factors and interactions play a role. Even within such recurrent circulation patterns, the 2018 heatwave in a factual world was still very unlikely. Moreover, by combining the storyline and flow-analogue approaches, we are not only able to project heatwaves under future warmer scenarios but also to assess their likelihood. This synthesis allows us to state that the

projected heatwaves in future scenarios are not as rare as the 2018 event was in the present climate, highlighting a critical shift in environmental risks as global temperatures rise. To conclude, combining the spectrally nudged storylines with the flow-analogue method provides a complementary physical-statistical framework for extreme event attribution that simultaneously explores the thermodynamic role of anthropogenic global warming, potential dynamical trends, and the changing likelihood of event occurrence, aiming to enhance communication and adaptation of climate extremes.

#### 6 Outlook





Further research using this combined methodology could be extended to other types of extreme events shaped by specific dynamical conditions, including cold spells, compound events such as heat/drought, or heavy precipitation and flood episodes. For events related to precipitation, higher-resolution datasets are recommended to more accurately capture the relevant processes in both storyline simulations and large ensembles. A limiting factor are the challenges models face in reproducing precipitation events, which may be overcome in the future by implementing AI-enhanced simulations of very high resolution. Additional constraints, similar to those used in Bayesian event attribution, could also be introduced to refine the event definition. For example, attributing not just a heatwave with a 2 °C anomaly under a given circulation pattern, but a heatwave accompanied by concurrent soil moisture deficits in compound events. This would allow for a more precise attribution of the event of interest. Moreover, causal inference could be explored within this framework by identifying, through the flow-analogue approach, the specific conditions under which the projected storylines tend to occur at different levels of global warming.

Code and data availability. All custom codes are implementations of standard methods in Python. The code used to produce the figures in this paper is available from the corresponding author upon request. The MPI-ESM-LR Grand Ensemble dataset is publicly available at https://esgf-data.dkrz.de/search/cmip6-dkrz/?mip\_era=CMIP6&activity\_id=RFMIP&institution\_id=MPI-M&source\_id=MPI-ESM1-2-LR. The ERA5 reanalysis dataset is available at the Copernicus Climate Data Store https://climate.copernicus.eu/. The global spectrally nudged storyline dataset is available upon request.

Author contributions. D.L.F.F. designed the study, methodology, performed analysis, data processing, visualization, wrote the first draft of the manuscript, reviewing and editing. A.L. and J.G.P. contributed to methodology, supported result interpretation, manuscript reviewing, and editing. F.F. supervised the research, contributed to the study design, supported result interpretation, manuscript reviewing, and editing.

Competing interests. The authors declare no competing interests.

Acknowledgements. This research was supported by the German Federal Ministry of Education and Research (BMBF) research program ClimXtreme II: A5 DesAttHeat (Grants number: 01LP2322A, 01LP2332B) and the European Union H2020 Project "CLIMATE INTELLIGENCE Extreme events detection, attribution and adaptation design using machine learning (CLINT)" [101003876-CLINT]. JGP thanks the AXA Research Fund for support.

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
