# Peer review of "A combined storyline-statistical approach for conditional extreme event attribution"

_EGUsphere, 2025_

## Referee Comment (RC2)

The manuscript "A combined storyline-statistical approach for conditional extreme event attribution" by León-FonFay et al. sketches out a new approach for conditional event attribution and demonstrate it with the 2018 European heatwave. They highlight the shortcomings of highly conditioned storylines, and combine them with a less conditioned circulation analogue method. This approach makes it possible to consider forced dynamical trends, and to quantify the changes in probability of occurrence.

I think it's a very important task to address the shortcomings of the individual attribution methods and to produce robust and interpretable attribution metrics that cover a variety of aspects, such as changing risks and intensities simultaneously. This framework provides a new perspective to think about and communicate attribution results, and I recommend it for publication in Weather and Climate Dynamics. While I want to highlight the value of this perspective, I find that the example used in the manuscript shows limitations of the approach which are not sufficiently discussed here. Therefore, I recommend major revisions which address the following points:

Major comment:
1) My main concern is around the interpretation of the results shown in Fig. 6 and 7. When the individual results from the storyline approach and the analogues are brought together, a new pattern emerges which provides the backbone of this new perspective: While the specific 2018 circulation was very extreme compared to the factual analogues (only 1.6% exceeding the event magnitude), the 2018 pattern becomes less extreme compared to the analogues with increasing GWL. We see this in Fig. 6 by the increasing fraction of the tails above the storyline values. Consequentially, the star markers in Fig, 7b don't align vertically as one might expect. This is a bold statement, and I think it should be part of the study to dissect the possible reasons behind this phenomenon, since this structure of the results is what makes up the novelty of the approach.
   • I can imagine possible physical explanations for this phenomenon: 2018 stood out not just by the circulation during the HW but also by the preceding precipitation anomaly. This led to dryer soils during the HW which exacerbated the impacts of the 2018 circulation. Due to the increasing thermodynamic drying of the future simulations, the role of the circulation intensity becomes less important, as more and more of the analogues also have drought conditions. You could test this by comparing the temperature anomaly of the 2018 circulation to the other summer temperature anomalies within each storyline scenario. If there is such an underlying physical explanation, the temperature anomalies of the 2018 circulation would move closer to the mean with an increasing GWL.
   • However, I think there might be a different explanation for the observed pattern, as I don't agree with the statement that the "approaches are physically consistent, as they rely on the same model" (L332). While the models share the atmospheric component, they differ in the ocean (fully coupled, vs AMIP based on observations scaled to warming patterns from large ensemble), and more importantly in the atmospheric dynamics (ECHAM6 vs nudged-to-NCEP). Small differences in climatology or warming rate between storylines and fully coupled grand ensemble would translate to a shift in the slope of the stars in Fig. 7b. I think

this is important to mention (or you could compare the ECHAM_SN climatology from (Schubert-Frisius et al., 2017) to the climatology in the grand ensemble?).

2) Selection of the analogs: From how I understand your approach, the main idea is to loosen the conditioning around a storyline in order to consider 'similar' events as well. However, it is not trivial to me to what extent the definition of similar will influence the outcome of this analysis. My impression from (Rousi et al., 2023) is, that the dynamical conditions that caused the 2018 HW were quite exceptional. It's understandable that these conditions must be loosened, but while your framework relies heavily on the concept of 'similar' circulation analogues, you set the definition in a way that defines the 2018 circulation as a 1-in-4 years event. Resulting from this, 83.5% of the analogues don't produce a HW under the factual climate. I'm not arguing that these conditions need to be stricter, but I would like to see a discussion on how the strictness in the conditioning of the analogues impacts a) the quantitative results, and b) how much the interpretation of the results hinges on the concept of 'similarity'.
I also agree with point 2b by reviewer #1, and think that a conditioning on the temporal pattern would pull away some weight from finding analogues with the right intensity (As seen in Fig. S4 the analogues are a bit on the weak side. For me, it would have helped to see some examples of the analogues and their related temperature patterns).

3) Forced circulation changes: The approach relies on a correct representation of P(D) by the grand ensemble. The fact that this is a topic of high uncertainty is one of the main reasons for using the storyline approach. While (Vautard et al., 2023) don't go into a separation of forced and unforced dynamical trends, their results suggest, that no member of the MPI-ESM GE is able to reproduce the trends in European circulation patterns related to summer heatwaves. Since the advantage of the proposed framework is its consideration of circulation changes, I think it should be discussed what we can realistically expect from CMIP6 models (Shaw et al., 2024).

To summarize: While there are currently limitations to the feasibility of the proposed approach, which are mostly based on the physical consistency between the two elements, I don't think that these constraints devalue the usability of this framework.

Minor comments:
• It looks like the model output and ERA5 have a 12h offset (while they both refer to the daily means). You could correct this to make the plot look nicer.
• The reference by (van Garderen et al., 2021) has the wrong year.
• The event definition (blue box in Fig. 2) is based on the impacts of the 2018 circulation, but the key point is to loosen the circulation restrictions, thereby including circulation analogues with different spatial temperature patterns. So I wonder if it would make sense to use a bigger box for the second part of the analysis?

Best regards,
Istvan Dunkl

References:

van Garderen, L., Feser, F., and Shepherd, T. G.: A methodology for attributing the role of climate change in extreme events: a global spectrally nudged storyline, Natural Hazards and Earth System Sciences, 21, 171–186, https://doi.org/10.5194/nhess-21-171-2021, 2021.

Rousi, E., Fink, A. H., Andersen, L. S., Becker, F. N., Beobide-Arsuaga, G., Breil, M., Cozzi, G., Heinke, J., Jach, L., Niermann, D., Petrovic, D., Richling, A., Riebold, J., Steidl, S., Suarez-Gutierrez, L., Tradowsky, J. S., Coumou, D., Düsterhus, A., Ellsäßer, F., Fragkoulidis, G., Gliksman, D., Handorf, D., Haustein, K., Kornhuber, K., Kunstmann, H., Pinto, J. G., Warrach-Sagi, K., and Xoplaki, E.: The extremely hot and dry 2018 summer in central and northern Europe from a multi-faceted weather and climate perspective, Natural Hazards and Earth System Sciences, 23, 1699–1718, https://doi.org/10.5194/nhess-23-1699-2023, 2023.

Schubert-Frisius, M., Feser, F., Storch, H. von, and Rast, S.: Optimal Spectral Nudging for Global Dynamic Downscaling, Monthly Weather Review, 145, 909–927, https://doi.org/10.1175/MWR-D-16-0036.1, 2017.

Shaw, T. A., Arblaster, J. M., Birner, T., Butler, A. H., Domeisen, D. I. V., Garfinkel, C. I., Garny, H., Grise, K. M., and Karpechko, A. Yu.: Emerging Climate Change Signals in Atmospheric Circulation, AGU Advances, 5, e2024AV001297, https://doi.org/10.1029/2024AV001297, 2024.

Vautard, R., Cattiaux, J., Happé, T., Singh, J., Bonnet, R., Cassou, C., Coumou, D., D'Andrea, F., Faranda, D., Fischer, E., Ribes, A., Sippel, S., and Yiou, P.: Heat extremes in Western Europe increasing faster than simulated due to atmospheric circulation trends, Nat Commun, 14, 6803, https://doi.org/10.1038/s41467-023-42143-3, 2023.

---

## Author Comment (AC1)

**Manuscript egusphere-2025-4976** *"A combined storyline-statistical approach for conditional extreme event attribution"*

**Response to Reviewer 1**

The work of León-Fonfay et al. proposes a framework to combine two conditional attribution methods of extreme events, one based on nudged simulations and the other one on flow analogues. While the first one gives thermodynamic changes for a dynamics following very closely the actual event, the other one follows more loosely the event but is able to assess conditional probabilities. The authors propose to combine the two to recover probabilities associated with the nudged approach. The method is illustrated by applying it to the 2018 Central European heatwave.

The paper is well-written and proposes an interesting method for improving results on an important topic of attribution. I especially appreciated the effort to find analogues in a large ensemble, which clearly enhances the quality of the results with this method. It is also a very worthy point to compare and combine attribution methods. It is therefore, in my opinion, suited for publication in WCD. I found the results obtained by the authors really interesting but also quite surprising and puzzling. I am not sure I agree at this point with their interpretation, especially on the fact that events such as the 2018 heatwave become much more likely in their own climate, including after conditioning on the same atmospheric dynamics. It seems to me that it may as well illustrate issues of the nudging method. I think this point needs more investigation to give an idea why the same dynamics, while being correctly recovered by the analogues, can give rise to much more intense events in the future. I therefore recommend major revisions. Please find below the details of my comments.

**Authors' response:** We appreciate the recognition of the contribution of combining storylines and flow-analogues for conditional attribution, and we agree that several of the points raised help clarify important methodological aspects and improve the outcome of our study. Below, we address each comment in detail and indicate how we have revised the manuscript accordingly.

**Major comments**

1. L125: The storylines simulation are nudged using NCEP reanalysis but you use ERA5 reanalysis to find the analogues. This seems like a potential issue to me. I do not expect the two reanalyses to have major differences but there could still be some. Especially the discrepancies in the temporal evolution in Fig 2c between ERA5 and the factual storyline do not seem so small. Please discuss that thoroughly or change the analogues analysis towards using NCEP data for the event.

   **Authors' response:**

   We thank the reviewer for noting this point. We chose ERA5 for analogue identification because of its higher resolution and better representation of European large-scale circulation and therefore a higher chance of obtaining an analogue

catalogue of high quality. To address this concern, we now include a comparison between ERA5 and NCEP anomalies for the event's circulation pattern (Supplementary Fig. S5) and the comparison to NCEP daily maximum temperatures in Fig.2c. Even though the storylines are nudged to NCEP, they get closer in temperature to ERA5, presumably due to the difference in resolutions between NCEP and ECHAM. The according inclusion of smaller-scale processes in the ECHAM simulations improve the output fields. NCEP resolution: T63 ~209km, ERA5 resolution: ~31km, ECHAM resolution: T255 ~ 60km.

[Figure]

**Figure S5. Analogues' ensemble mean time-averaged GPH@500hPa anomaly compared to reanalysis (ERA5 & NCEP) mean flow pattern...**

[Figure]

**Figure 2. The 2018 Central European heatwave...**

2. Sec 3.2: I think the way analogs are found needs some precision because this may be important for interpreting your results correctly.

   a. How are the GPH anomalies detrended precisely? Please note that using grid point anomalies and detrending by grid points break the horizontal gradients and therefore the closeness between the winds of the event and the analogues (because of the geostrophic balance). In particular the model and ERA5 may not have the same climatology and therefore comparing anomalies may not compare patterns that are actually physically similar. Similarly, the climatology likely changes between the different periods in the model (both in mean and variance). Finally, the climatology of ERA5 is problematic to compute: how did you estimate the forced component in GPH ?

   **Authors' response:**
   We agree that the use of a grid point-wise detrending does not fully conserve horizontal gradients. Moreover, we understand the concern that the climatological state may differ between ERA5 and the MPI-ESM model. This is why we tested how much the climatological states differ between ERA5 and the MPI-ESM ensemble mean for present day conditions. Overall, the summer-mean Z500 fields look very similar between MPI-ESM and ERA5 (Fig. R1a,b). The differences in the mean state (Fig. R1d) are substantially smaller than the average Z500 anomalies of the 2018 HW analogue cases (depicted in Figure 5

in the main article). We are therefore certain that most of the analogue cases should display very similar large-scale flow conditions irrespective of the minor differences in the climatological background between ERA5 and MPI-ESM.

In addition, we also show the difference between the 3K warming level and present-day conditions for the MPI-ESM ensemble (Fig. R1c). To focus on possible differences in the overall flow pattern, we remove the global warming signal by subtracting the area-averaged difference over a box from 20°W to 40°E and 30°N to 80°N. The differences between the +3K and present-day ensemble mean (Fig. R1e) then become very small. This assures us that most analogue cases will feature a similar flow regardless of the background climate.

Finally, we want to point out that the analogue method is not designed to confine the flow very strictly to the observed event. This is what the spectrally-nudged storyline are specifically designed for. With the analogues, we want to allow for consistent but still a rather broad spectrum of outcomes. While we search for similar patterns in Z500, the flow evolution in each given analogue case will definitely show some distinct deviations in the evolution of the large-scale flow. Moreover, similar Z500 patterns may also feature substantial differences in the near-surface flow and other quantities that may affect the evolution of temperature anomalies.

[Figure]

**Fig. R1: A comparison of the summer 500hPa geopotential climatology between ERA5 and the MPI-ESM large ensemble**. The left column shows the climatological 500hPa geopotential average over the summer months June, July and August for a) ERA5 in the period 1985-2014, b) the ensemble-mean of the CMIP6 historical simulation from MPI-ESM for the simulated years 1985-2014 and c) the MPI-ESM ensemble mean for time slices representing a 3K warming level. d) shows the difference between the MPI-ESM and ERA5 for the current climate (1985-2014). e) features a comparison between Z500 fields between the 3K warming level and present-day conditions for the MPI-ESM ensemble. To remove the mostly uniform global warming signal in the geopotential, we have subtracted the difference in the field mean over a box from 20°W to 40°E and 30°N to 80°N.

b.   Am I right if I say that you compute the euclidean distance over 5-days or are you averaging first over these 5 days ? The first option seems problematic to me because it vastly increases the dimension of the space in which you are looking for nearest neighbors and therefore risks finding worse analogs (in other words, it is way more difficult to find events which have the same dynamics over 5 days than over 1 day).

**Authors' response:**

We designed the analogue detection algorithm following our colleague (and flow-analogue method developer (Zorita and von Storch, 1999)) Eduardo Zorita's advice to account for the sequence of the pattern while keeping good quality analogues. Including several days indeed reduces the number of available analogues, but it allows us to capture the temporal evolution of the circulation pattern. A 5-day sequence offers a good balance: it preserves the event's chronology while still returning a sufficient sample size. Using a 5-day mean Euclidean distance instead increases the number of analogues, but at the expense of losing the sequence information that characterizes the event. We also acknowledge that a 5-day sequence may result in fewer (and potentially poorer) analogues, which is why we apply a second quality filter based on mean spatial correlation to retain only high-quality matches. Finally, we can allow ourselves to use these two thresholds due to the very large dataset we have. In total we have 92000 days available per global warming level (92days (June-August) x 20 years x 50 members) for the analogue detection.

We did a brief evaluation on the use of a 5-day mean instead of the 5-day sequence (see Fig. R2). The associated temperature anomalies in Central Europe do not change, but we get a larger sample of analogues from the 5-day mean approach, at the cost of containing less information in the pattern evolution. We therefore keep the 5-day sequence, since it provides a large enough sample based on more detailed information of the pattern.

[Figure]

**Fig. R2. Analogue detection conditioning.** The blue distribution corresponds to the analogues identified in the MPI-ESM GE using a 5-days sequence as a reference pattern. The red distribution corresponds to the analogues identified using a 5-day mean of the reference pattern. N shows the number of analogues.

    c. I find the region to look for the analogues slightly strange: you are interested in temperature in Germany, therefore the anticyclone (which is local) over Germany should be your target field. I am not really

convinced about using a box so extended to the West. In particular, it is possible that in your analogs you have events with a low pressure in the south of Iceland but few high pressure over Germany. That being said I am not sure it would be a major change to the results because you already have a large part of the domain included.

**Authors' response:**

The peak of the heatwave event in July 2018 was first characterized by a dipole pattern of lower than normal pressure over the North Atlantic and elevated pressure over the European continent (see figure S1 in the Supplement). Later on, the cyclone over the Atlantic receded northwards, and the high pressure anomalies extended westwards. To fully capture this evolution of the large-scale flow, we found it appropriate to extend the region for analogue detection to the west.

We have included in the text a new section in Supplementary:

**Supplementary Section 1.2: Region for analogue detection**

*"In Fig. S3 we illustrate the outcome of using the 2-step analogue detection in: a large region capturing the synoptic domain (box 1), the westward shifted domain used to include the dipole pattern (box 2), and a smaller region surrounding Central Europe (box 3). In general, using box 1 and box 2 allows us to account for large-scale gradients in the geopotential field. In combination with the inclusion of multiple days, we are thereby able to narrow down the selection of analogues to those with a very similar evolution of the large-scale flow, also with regard to the advection of air masses.*

*In Fig. S3b, the temperature anomalies in Central Europe indicate that the distribution remains quite comparable between box 1 and box 2, with the disadvantage of box 1 having a highly reduced number of analogues. The small region (box 3) results in selecting many more cases that also feature high geopotential over Central Europe, but with possibly quite different developments of the synoptic-scale flow and advection of air masses. The drawback of box 3 is that it captures any high pressure system over Central Europe, neglecting information about the characteristics of the blocking system. Therefore, we find that box 2 is the most suitable for our study."*

[Figure]

**Fig. S3. Region for analogue detection.** a) Geographical domains used for the analogue detection: the Central European target region (blue), the larger synoptic domain (box1: yellow), used synoptic domain box (box2: green), and the surrounding Central-European subregion (box3: magenta). b) Distribution function of t2m anomalies (TX-TX95) associated with the analogue catalogue for each region, N is the number of analogues detected (see label), illustrating how the choice of analogue domain influences the temperature distribution in Central Europe.

d.  I am not sure I understand your two steps procedure to find the analogs: why first selecting the N closest events if after that you impose again a hard threshold on the spatial correlation? Why not directly using the spatial correlation ? By the way I think using a hard threshold of spatial correlation requires some thinking: the correlation for each day is scaled by the spatial variance of that day, which would be different from one pattern to another. This implies that events with a similar absolute distance may have different spatial correlation. This has important implications when estimating P(D). I would advise rather using a similar absolute euclidean distance for each period because this has a straightforward interpretation: this is a ball around your event of interest in the phase space and counting the number of events in the ball gives you an estimate of the probability to fall into this ball, i.e. P(D). Estimating the sensitivity of your results to this threshold would also be good given that this is one of your main results.

**Authors' response:** As discussed in point 2.b, the two-step procedure is designed to capture both the **temporal sequence** of the circulation pattern and its **mean spatial structure**. The first step (Euclidean distance) ensures similarity in the 5-day evolution of the pattern, while the second step (spatial correlation) ensures that only high-quality analogues with a comparable spatial structure are retained. This filter is strict but due to the large dataset we have,

it returns a sufficiently large set of **high-quality analogues**. We have added in the text:

**Lines 179-180:** *"Conceptually, our approach asks: given this circulation pattern with this sequence and mean structure, what is the spectrum of heatwaves that can occur?"*

3. Sec 3.4: I am not sure I see the interest of computing the probability P(E,D). I think there are two interesting probabilities here: either P(E|D) which is the probability to observe E given the pattern D — the statement in L192 that "the probability of an event of magnitude M, under the circulation pattern of interest D, in a given global warming level (GWL) will be defined as P(E,D)" is not true I think —, or P(E) which is the unconditional probability to observe your event. Because P(E) = P(D)P(E|D) + P(D̶)̶P̶(̶E̶|D) you could compute this absolute probability, but this requires to know P(E|~D) which is a probability that can change with climate change. Thus I am not sure that an absence of change in P(D) can be so straightforwardly interpreted. Finally, you mention in L195 that you fit a GEV to the analogs conditional distribution and I see no reason to proceed as such: the use of GEV in extreme value theory is justified by the extremal theorems that state that the limit of block maxima distributions should follow a GEV. You are not using block maxima so it is not clear why your conditional distributions should follow a GEV (Figure 6 in particular does not plead in your favor). In a similar context Noyelle et al. (2025) used for example a skew normal distribution. Given the results of your Figure 6 and the large number of analogs you have, why not simply computing the probability empirically by counting the number of analogs above your target ?.

**Authors' response:** We have rephrased this statement in Section 3.4 of the manuscript.

**Lines 196-211:**

*"In our study, we are interested in the probability that a heatwave of magnitude M co-occurs with the circulation pattern of interest (flow-analogues) in the dataset representing a given global warming level. To calculate this probability, we use the conditional probability definition P(E,D) = P(D)P(E|D), where E is the heatwave of interest of magnitude M, and D is the circulation pattern related to the event. Each term provides valuable information in our assessment:*

- *P(D): The probability that we find the circulation pattern in the dataset. (addresses changes in dynamics with global warming). Example: D= "blocking over Western Europe". If Blocking occurs 25% of the years P(D) = 0.25*
- *P(E|D): Is the probability of E giving that circulation pattern D is present. Example: E = "Heatwave in Germany of magnitude M". When blocking occurs (analogues), 10% of them surpass the event's E magnitude, then*

*P(E|D)=0.1. This is the probability that a heatwave exceeds the magnitude M within the flow-analogues subset.*

- *P(E,D): Represents the total fraction in the dataset on which both the circulation pattern and the extreme event co-occur. P(E,D) = P(E|D) P(D) = 0.25 x 0.1 = 0.025. Following this example, there is a 2.5% probability that Germany experiences a heatwave and a blocking simultaneously that resemble the conditions we are interested in for this specific global warming level. For simplicity, we refer to this probability as $P_{M,GWL}$. "*

On the other side, P(E) is also interesting, and could be calculated by counting the number of events exceeding the target in the dataset without conditioning on the circulation, but this one would include all kinds of dynamics. Finally, we agree that counting the number of elements beyond a threshold would be enough, but we fitted our flow analogues distributions to account for uncertainty ranges using bootstrap-resampling and to extrapolate magnitudes and return periods that go beyond the available ones in our dataset.

We also included the empirical data in Fig. 7 to show that GEV is a good fit for extrapolation.

[Figure]

**Figure 7. Heatwave intensity's return periods...**

4. The most surprising result of this paper is that even when compared to their own climate, the probability to observe an event like the 2018 one increases vastly. This result is obtained by comparing (i) the intensity of the 2018 event obtained by the nudged storyline method in different climates and (ii) the probability to observe such an intensity in the new climate conditional on having the same large scale atmospheric pattern (i.e. using the analogs). From this, Figure 6 clearly shows that this latter probability increases in future climates because of an increase in the variance of the conditional distribution. This increase in the variance is expected and likely comes mainly from the strengthened feedback with soil moisture in a hotter, drier summer climate in

Europe (e.g. Fischer et al. 2012), because, as shown by the Authors', the intensity of the analogs GPH pattern (and therefore by extension of advective and adiabatic contributions of heatwaves) does not seem to change. On the other hand the change in the mean seems to be mostly linear with GW, which is also reflected in your nudged simulations (Fig. 4b). The Authors' then interpret this change as a real change in probability for such an event as in 2018 and this is where I am not convinced. I think the way the nudging simulations are designed strongly constrains the soil moisture component and therefore that the method proposed here essentially compares a distribution with fixed soil moisture (nudging) and one with varying soil moisture (analogues), which inflates artificially the probability obtained. To detail: the nudged simulations run for 10 years with the winds observed over the period 2015-2025 in climates with different warming levels. This implies that the soil moisture component is driven by different thermodynamical forcings but similar dynamical ones, which is a strong constraint given that rain and evapotranspiration are strongly dynamically driven variables. As such, the soil moisture distribution in the nudged simulations — and especially at the beginning of the 2018 event — are likely different from the one of the MPI LE, from where the analogues are taken. For a fair comparison, the winds should be nudged for a realistic distribution of soil moisture (i.e. with respect to the soil moisture distribution of the LE) some weeks before the event or the analogues should be taken conditional on a soil moisture distribution similar to the nudged simulation (which would likely decrease the variance in Figure 6 and therefore the change in probability). Here you assume that the change in intensity given by the nudging is correct and infer from that the change in probability from the analogues. I think one could do exactly the inverse: given the probability in the factual period, take similarly unlikely events in other periods and infer from that the change in intensity, which would rather show that the nudging underestimate the change in the intensity of the event. Currently, I am personally more convinced by this interpretation than by yours.

To take a step back and summarize, my main issue with the current reasoning presented is that the Authors' assume that nudging the winds in a different climate only conditions the dynamical component: this is not true in general because global warming has both dynamical and thermodynamical components that are intrinsically linked. One main example is the thermal wind relationship: by imposing the winds you also impose the thermal gradients, despite the fact that those thermal gradients do change with climate change. In other words, the nudged simulations impose an atmospheric dynamic that is physically not realistic — at least at long time scales.

**Authors' response:**

Thank you for the detailed comment. We see some main points to be addressed:

**(1) Influence of soil moisture in the storylines and analogues.**

We agree that soil-moisture feedbacks could potentially be a source of the increased variance in future climates as seen in Figure 6 (we added the distribution width in the label) (Fischer et al., 2012). But of course other factors like changed sea surface temperatures and greenhouse gases will also have an effect. We also agree that constraining the upper-level horizontal flow has a strong effect on precipitation and evapotranspiration but is of course a prerequisite for conditional attribution using spectrally nudged storylines in order to be able to attribute historic extreme events. However, our analyses do not support the interpretation that the nudged simulations impose a fixed or unrealistic soil-moisture state. The nudging in our simulations is only applied for vorticity and divergence at very large spatial scales above 750hPa, letting all other variables and processes close to the surface evolve freely.

- Figure 8a shows that soil moisture evolves differently across warming levels for the storylines and is not constrained by the nudging.

- Figure 8b demonstrates that the analogues at each warming level occupy temperature(T)-soil moisture(SM) states similar to those in the corresponding storylines and that also their number of extreme heatwaves increases in their own warming levels when soil moisture is included as an additional condition in the analogue attribution  (area of bivariate (SM - T) threshold exceedances shown as darker dots in Fig. 8b).

- If one conditions the temperature (T) distributions on the soil moisture (SM) of a storyline (i.e. p(SM=60, T) ), the variance is largely reduced, but the sample size is too small (~30 elements) to perform significant attribution.

This confirms that the approaches are compared between two consistent conditional states under the same circulation pattern, where the analogues lead to a wider range of possible events deriving from the same pattern due to other underlying processes.

We included this discussion and the new Figure 8 in the manuscript (see below). Furthermore, we included the width of the analogues' distributions in Figure 6.

[Figure]

**Figure 8. Soil moisture.** *a)* Time series of daily soil moisture over Central Europe for each storyline. Solid lines show the ensemble mean, and shaded bands indicate the ±2std range across the five ensemble members for each storyline. *b)* Bivariate distribution of analogue events, displaying soil moisture (y-axis) versus temperature anomaly (x-axis). Colored lines show the soil moisture-temperature values from each storyline. Scatter points represent analogue-derived values, with the darker points indicating analogue events that exceed the soil moisture and temperature thresholds of the corresponding storyline.

[Figure]

**Figure 6. Temperature anomaly distribution for best flow-analogues of the 2018 HW...**

**(2) Interpretation of the probability increase**

There could be a number of reasons for the increase in probability in future climates. These could be related to, for instance:

- Surface processes, sea surface temperature, aerosol, or greenhouse gas changes differing between approaches.
- In addition, the model set ups for storylines, and the GE may not be identical (e.g. atmospheric only versus coupled model) despite sharing the same atmospheric model.
- Besides, the storylines depict climate change in a relatively simple and more conservative way, changing only SSTs and greenhouse gases between storylines, whereas MPI-ESM includes also other processes such as aerosol changes. However, this is due to the definition of the storylines as we did not want to include variables whose change in future climate is uncertain to be caused by human impact or variables which change very differently in scientific studies.

Nonetheless, both attribution methods are conditioned on the observed dynamics and were made as comparable as possible. The fact that the upper tail of the global warming levels in MPI-ESM increases more prominently – maybe because of soil moisture feedback- does not imply that the approaches disagree. Instead, it reflects that MPI-ESM leads to a larger spectrum of possible heatwaves developing under similar dynamics. The storyline approach provides a representation of how this single event would change in response to anthropogenic thermodynamic forcing (SSTs and GHGs). The flow analogue approach includes a wider range of heatwaves developing under similar dynamics but different interacting processes, including more extreme heatwaves probably resulting from a reduced soil moisture availability.

Based on our new Fig. 8 (Fig. R3) we conclude that both analogs and storylines support the statement that future events with a similar blocking system to the 2018 heatwave become more frequent and extreme in future climates than projected by the storyline approach. We rephrased our conclusions to account for the still unknown reasons for the increased probabilities, the detection of these is unfortunately beyond the scope of this study.

**(3) Physical realism and dynamical–thermodynamical consistency**

We are not sure if we totally understand your comment that the nudged simulations are not physically realistic on long time scales. Is this meant to refer specifically to future climate states? For present time the nudged simulations are of course perfectly realistic. The purpose of the storylines is to ask: *"What would this specific circulation have produced under a different thermodynamic state?".* Keeping the same dynamics is therefore essential for this study.

Of course there will be dynamical changes in future climate states. But as we could show in Figure 6, the number of dynamical situations similar to the one that happened for the heatwave 2018 reflected by the analogues of the MPI-ESM-GE is constant, even for the most extreme states. We thus conclude that the nudged simulations are not unrealistic, even for future climate states.

To reply in full to point 4, in the manuscript we included this discussion in the new **Section 4.3.1 Increased probability of extreme heatwaves in warmer worlds** and lowered down the tone of our conclusions related to the result obtained through the combined approach:

**Lines 5-10**

*"... Despite no detected changes in terms of atmospheric blocking, the flow-analogue approach further indicates that heatwaves exceeding the storyline-projected intensities become more frequent and extreme at their corresponding warming levels than the factual 2018 event was under present conditions. Specifically, the 2018 heatwave, with an intensity of 2.2 °C and a return period of 1-in-277-years today, becomes a 6.6 °C event with a 1-in-26-year probability in a +4K world. **This behavior revealed the importance of other physical mechanisms and interactions beyond the atmospheric circulation pattern and thermodynamic conditions influencing the occurrence and intensification of heatwaves.**"*

**Lines 351– 386:**

" Section 4.3.1 Increased probability of extreme heatwaves in warmer worlds

*We hypothesize that this increase in extreme heatwaves could come from a combination of factors: 1) the conservative definition of global warming in the storyline approach (which only imposes changes in SST and GHGs, as they are certain to have a human-induced contribution) restricts the heatwave intensities. 2) the integration of more complex interactions in the MPI-ESM GE (coupled earth system model) in the flow analogues compared to the ones represented by the storylines simulated with ECHAM (atmospheric model with an integrated land component JSBACH). 3) The role of soil moisture as a source of variance in the temperature distributions for future warming levels (Fischer et al., 2012).*

*The 2018 heatwave was very extreme, not only because of its circulation, but also because of its exceptionally preceding dry conditions (Rousi et al., 2023). In Figure 8, we explore the role of soil moisture in both approaches. In Fig. 8(a), we see how soil moisture is also affected by global warming, decreasing at a faster rate in future levels. The counterfactual storyline had a mean soil moisture of 62.9 kg/m2 during the main heatwave event, while in a +4K storyline, the soil moisture dropped to 54.9 kg/m2. In Figure 8(b), we show the bivariate distribution of soil moisture (SM) and temperature anomaly (T) associated with the flow analogues. The lines in this plot show the corresponding magnitudes in the storyline approach per global warming level. The bivariate comparison demonstrates that the analogues at each warming level occupy temperature(T)-soil moisture(SM) states similar to those in the corresponding storylines and that also their number of extreme heatwaves increases in their own warming levels when soil moisture is included as an additional condition in the*

*analogue attribution (area of bivariate (SM - T) threshold exceedance shown as darker dots in Fig. 8(b)). Here, we can provide evidence that despite having similar atmospheric circulation patterns, the analogues do portray a larger spectrum of events that could emerge due to other underlying conditions, like enhanced soil moisture deficit. It is possible that due to the increasing thermodynamic drying of the future simulations, the role of the atmospheric circulation intensity becomes less important, as more of the analogues also have drought conditions.*

*Furthermore, the increase of extreme heatwaves in the upper tail of the temperature (Figure 6) and bivariate SM-T (Figure 8) distributions does not come only from the shift in the mean of the distributions towards warmer temperatures due to global warming, but there is also an increase the variance in future levels (see label Figure 6), which could be associated to a strengthened feedback with soil moisture in the analogue catalogue. If one could condition the temperature distributions on the soil moisture of a storyline (i.e. p(SM=60, T) ), the variance would be largely reduced. This would lead unfortunately also to a smaller sample size, too small to perform significant attribution.*

*In sum, both attribution methods are conditioned on the observed dynamics and were made as comparable as possible. The fact that the upper tail of the global warming levels in MPI-ESM increases more prominently does not imply that the approaches disagree. The storyline approach provides a representation of how this single event would change in response to anthropogenic thermodynamic forcing (SSTs and GHGs). The flow analogue approach includes a wider range of heatwaves developing under similar dynamics but different interacting processes, including more extreme heatwaves, probably resulting from a reduced soil moisture availability. Even though these results may be model-dependent, it is worth emphasizing that both approaches rely on the same atmospheric model. The storylines are simulated with ECHAM6, the atmospheric component of the MPI-ESM model, while the flow-analogues are extracted from scenario simulations of MPI-ESM. Including additional models for comparison in future work could provide more detailed insights on this matter. "*

***Lines 408 –415:***

*"Warmer storylines of the 2018 heatwave are not as rare at their corresponding level of global warming, as the 2018 heatwave was in the present. While the heatwave intensities increase linearly per degree of global warming, their frequency exponentially increases. Specifically, the observed 2018 heatwave had a 2.2 °C intensity with a return period of 1-in-277 years in the present time, intensifying to 6.6 °C with a return period of 1-in-26 years in a +4K world. **The reason for the increased intensification in future climates is subject to further studies. So far, we assume that these differences could emerge from the conservative representation of global warming in the storyline approach, soil-feedback mechanisms enhanced in the MPI-ESM grand ensemble that leads to a larger variance with global warming, and other processes accounted in the coupled model"***

**Lines 459 – 462:**

*"The major outcome of this study is the combination of two conditional attribution methods, namely storylines and analogues. In this way we can provide information how a specific historic extreme event would have occurred without anthropogenic warming and how it might behave for future climate states in combination with statistics like probabilities of occurrence for classes of similar extremes..."*

**Lines 467 – 473:**

*"In Section 4.3.1 we discuss the possible reasons for this increase in probability. According to Feser et al. (2024), the spectrally nudged storyline approach is conservative as it represents the influence of anthropogenic global warming solely through information stored in sea surface temperature and greenhouse gases, while other variables, which are likely also influenced by human activity -like aerosols- are not taken into account, representing climate change more cautiously.* **This may restrict heatwave intensity, which accompanied by the role of soil moisture in temperature variance and other unknown processes coming from the coupled model may explain the increase in the upper tail distribution of the heatwaves detected in the analogue approach, compared to the simulated storylines."**

**Lines 483-489:**

**"Naturally, it is not only the atmospheric circulation pattern and global warming that define the characteristics of a heatwave.** *For such reason, not all flow-analogues evolve into a heatwave,* **and others get much more intense than the observed one.** *Many other factors and interactions play a role,* **like preceding drought conditions.** *Moreover, by combining the storyline and flow-analogue approaches, we are not only able to project heatwaves under future warmer scenarios but also to assess their likelihood.* **This synthesis allows us to state that future events with a similar blocking system to the 2018 heatwave might become more extreme and more frequent in future climates than the 2018 event was in the present***, highlighting a critical shift in environmental risks as global temperatures rise."*

**Minor comments**

1. L13: please note that cold waves have decreased worldwide

   **Authors' response:**

   *"The number of* **heat-related** *extreme weather and climate events has increased in recent decades"*

2. L59-61: it is not clear to me what the numbers are referring to precisely, would it be possible to give more precision of what are the events considered and what is the climatology here?

   **Authors' response:**

   **Lines 64-65:**

   *"... led to a 5.7 °C exceedance of the **1985-2014** climatological temperature over the Balkans, while in a +2K world, one could expect Poland to experience up to 7.6 °C above the **same** climatology"*

3. L73: "While this method is less suited to attributing the precise influence of anthropogenic warming on a single event, it provides robust estimates of the likelihood of similar events occurring under varying climate conditions" -> I am not sure this is true: the 'precise influence of anthropogenic warming on a single event' is not something that is well defined because a single event is single, i.e. it will never reproduce identically in the future, even in a stationary climate. All attribution methods require creating a class of events. This is also true for the nudging method: this is the class of events that have exactly the same upper-level winds in the future as the event observed. What is at stake when comparing attribution methods is precisely to know how the class of events defined gives a relevant answer to the question one wants to answer. In this sentence you seem to assume that you know which method is a priori better, which is really not sure I think.

   **Authors' response:**

   We did not want to imply that spectrally nudged storylines would be better than flow analogues. On the contrary, the whole article is about the added value of combining both methods as they deal with different aspects of attribution. Of course, a certain extreme event will never happen identically in the future. However, it is important to attribute a single extreme event as there is a lot of interest on how much it was influenced by anthropogenic warming, but also on how it could have theoretically evolved in a future climate. Especially if this extreme event had a large impact and is still anchored in collective memory. We do not agree with the statement that also for nudged simulations we are creating classes of events. The ensemble simulations are very similar to each other, which is not comparable to the difference between extremes when not using spectral nudging. So we can say that we are attributing the role of climate change for a single extreme event.

4. Sec 2.2: It was initially a little bit confusing to me what you call storylines in this section, because you seem to argue that a climatological simulation of a +2K world is a storyline. In the end you use the nudged simulations only for some days around your event, maybe you could then present the nudged simulations in a simpler way. Also, could you precise at which levels the winds were nudged ?

**Authors' response:**
**Lines 127-129:**
*"The spectral nudging of vorticity and divergence (nudged for wave numbers n<38, >750hPa) towards NCEP-NCAR reanalysis (Kalnay et al., 1996; Kistler et al.,2001) ensures that the large-scale weather pattern stays close to the observed one in all storylines"*

**Lines 132-134:**
*"As a result, all 5 storylines listed above are available for the same period, running from 2015 to the present day under different warming conditions. **This means that each storyline contains the historical events occurring globally during this time period (at least those reproducible at this resolution).**"*

5. L149-155: I think I disagree with the statements in this paragraph. I do not think it is true to say that the nudged storylines answer the question "what is the influence of anthropogenic global warming on the extreme event of study". I must say I think this question is not well posed and therefore cannot be answered without additional precisions: if you consider the phase space of the climate system and you take a point in this phase space (i.e. your extreme event), it is meaningless to ask how climate change changed this point. An event is an event, it does not mean anything to ask how it changed. You can ask how its probability changed (change in the stationary distribution for this point), how its intensity changed (change in the probability to observe a function of this point) etc but these questions always require creating a class of events, i.e. a set in the phase space. The discussion about different attribution methods is precisely how to define these classes so that they answer the question we want to answer. The nudged storylines method is just one way to define this class, and maybe not the best (see my major comment 4).

**Authors' response:**
These methodologies provide different information. Addressing the change in probability or intensity based on a set in the phase space is the usual probabilistic approach presented by Otto (2017). We don't aim to state that storylines are a better approach, but an alternative instead. The nudged storyline approach is not meant to redefine a class of events, but to provide a conditional causal experiment: it estimates how the same circulation pattern would manifest under different background thermodynamic states, communicating 'what if' this event would have happened in a pre-industrial or warmer world. This complements, rather than replaces, probabilistic attribution methods that assess changes in likelihood across event classes.

6. Fig 3 and L216: at these places and several other places you do not define precisely what is the climatological percentile that you compare the storylines

to. If it is the "local 95th percentile of the climatological daily maximum temperature" then I think the figure is wrong because the local percentile should be the percentile of the climate of each storyline. I guess you are comparing to the percentile in the factual climate, which is fine, but this needs to be precised so that the reader understands that most of the change comes from an increase in the mean. Same comment for the climatological 95th percentile in L286.
**Authors' response:** We have clarified it in the text.
*"Here we define intensity as the exceedance (in °C) of the local 95th percentile of the climatological (factual 1985-2014) daily maximum temperature (see Sect. 3.3)."*

7. Fig 4a: Is this a mix of all storylines ?
**Authors' response:** Yes, we've clarified it in the text:

**Lines 245-247:**
*"The main contribution of anthropogenic global warming to temperature increase over the region of interest can be evidenced in Fig. 4 (t2m increase per degree of global warming, computed as the linear trend across the storyline simulations)"*

8. L314: "In a factual world, the 2018 heatwave conditioned to such an atmospheric circulation pattern had a 1-in-277-years" -> I don't think this is correct, the conditional probability/return time is the one you give at the next line: 1/1.6%=62.5 years.
**Authors' response:** Thanks, we've clarified this point by expressing the probabilities in terms of their correspondent expressions.

**Lines 333-335:**
*"In a factual climate, roughly ~ 25% of years exhibit a similar blocking pattern (P(D)), and only ~ 1.6% of these analogues exceed the observed mean heatwave intensity of 2.2 °C (P(E | D)) (Fig. 6). Hence, in the factual period, the 2018 heatwave conditioned on this circulation pattern was a rare event, occurring with a probability of about 1-in-277 years (1 / $P_{2.2°C,F\ actual}$) according to the fitted value (Fig.7)..."*

References

Noyelle, R., Faranda, D., Robin, Y., Vrac, M., & Yiou, P. (2025). Attributing the occurrence and intensity of extreme events with the flow analogue method. Weather and Climate Dynamics, 6(3), 817-839. https://doi.org/10.5194/wcd-6-817-2025.

Fischer, E. M., Rajczak, J., & Schär, C. (2012). Changes in European summer temperature variability revisited. Geophysical Research Letters, 39(19). https://doi.org//10.1029/2012GL052730.

Otto, F. E. (2017). Attribution of weather and climate events. *Annual Review of Environment and Resources*, *42*, 627–646. https://doi.org/10.1146/annurev-environ-112621-083538

Rousi, E., Fink, A. H., Andersen, L. S., Becker, F. N., Beobide-Arsuaga, G., Breil, M., Cozzi, G., Heinke, J., Jach, L., Niermann, D., et al. (2023). The extremely hot and dry 2018 summer in central and northern Europe from a multi-faceted weather and climate perspective, *Natural Hazards and Earth System Sciences, 23,* 1699–1718. https://doi.org/10.5194/nhess-23-1699-2023.

Zorita, E., & von Storch, H. (1999). The Analog Method as a Simple Statistical Downscaling Technique: Comparison with More Complicated Methods. *Journal of Climate, 12(8)*, 2474-2489. https://doi.org/10.1175/1520-0442(1999)012<2474:TAMAAS>2.0.CO;2.

---

## Author Comment (AC2)

**Manuscript egusphere-2025-4976 *"A combined storyline-statistical approach for conditional extreme event attribution"***

**Response to Reviewer 2**

The manuscript "A combined storyline-statistical approach for conditional extreme event attribution" by León-FonFay et al. sketches out a new approach for conditional event attribution and demonstrate it with the 2018 European heatwave. They highlight the shortcomings of highly conditioned storylines, and combine them with a less conditioned circulation analogue method. This approach makes it possible to consider forced dynamical trends, and to quantify the changes in probability of occurrence.

I think it's a very important task to address the shortcomings of the individual attribution methods and to produce robust and interpretable attribution metrics that cover a variety of aspects, such as changing risks and intensities simultaneously. This framework provides a new perspective to think about and communicate attribution results, and I recommend it for publication in Weather and Climate Dynamics. While I want to highlight the value of this perspective, I find that the example used in the manuscript shows limitations of the approach which are not sufficiently discussed here. Therefore, I recommend major revisions which address the following points:

**Authors' Response:**

Dear colleague, thanks for recognizing the value of our study. Your comments have been constructive and helped with the improvement of the manuscript. In the following points, we describe in detail how we have implemented such comments in our study.

Major comment:

1) My main concern is around the interpretation of the results shown in Fig. 6 and 7. When the individual results from the storyline approach and the analogues are brought together, a new pattern emerges which provides the backbone of this new perspective: While the specific 2018 circulation was very extreme compared to the factual analogues (only 1.6% exceeding the event magnitude), the 2018 pattern becomes less extreme compared to the analogues with increasing GWL. We see this in Fig. 6 by the increasing fraction of the tails above the storyline values. Consequentially, the star markers in Fig, 7b don't align vertically as one might expect. This is a bold statement, and I think it should be part of the study to dissect the possible reasons behind this phenomenon, since this structure of the results is what makes up the novelty of the approach.

**Authors' Response:**

Thanks for observing this point. Reviewer 1 has also raised the same concern. In order to address this issue, we have: 1. Explored the role of soil moisture in both approaches, 2. Acknowledge the possible differences between approaches (the atmospheric model used in the storyline approach vs. the coupled model used in

the flow analogues), 3. Rephrased our conclusions to speculations of why we might face an increased probability in warmer worlds, but still stating that our results show that heatwaves with a similar dynamical conditions as the 2018 heatwave become more likely and more extreme due to still unknown mechanisms beyond the fixed circulation, increasing SSTs and GHGs with global warming.

In the manuscript we included this discussion in the new **Section 4.3.1 Increased probability of extreme heatwaves in warmer worlds** and lowered down the tone of our conclusions related to the result obtained through the combined approach:

**Lines 5-10**

*"... Despite no detected changes in terms of atmospheric blocking, the flow-analogue approach further indicates that heatwaves exceeding the storyline-projected intensities become more frequent and extreme at their corresponding warming levels than the factual 2018 event was under present conditions. Specifically, the 2018 heatwave, with an intensity of 2.2 °C and a return period of 1-in-277-years today, becomes a 6.6 °C event with a 1-in-26-year probability in a +4K world. **This behavior revealed the importance of other physical mechanisms and interactions beyond the atmospheric circulation pattern and thermodynamic conditions influencing the occurrence and intensification of heatwaves.**"*

**Lines 351– 386:**

" Section 4.3.1 Increased probability of extreme heatwaves in warmer worlds

*We hypothesize that this increase in extreme heatwaves could come from a combination of factors: 1) the conservative definition of global warming in the storyline approach (which only imposes changes in SST and GHGs, as they are certain to have a human-induced contribution) restricts the heatwave intensities. 2) the integration of more complex interactions in the MPI-ESM GE (coupled earth system model) in the flow analogues compared to the ones represented by the storylines simulated with ECHAM (atmospheric model with an integrated land component JSBACH). 3) The role of soil moisture as a source of variance in the temperature distributions for future warming levels (Fischer et al., 2012).*

[Figure]

**Figure 8. Soil moisture.** *a)* Time series of daily soil moisture over Central Europe for each storyline. Solid lines show the ensemble mean, and shaded bands indicate the ±2std range across the five ensemble members for each storyline. *b)* Bivariate distribution of analogue events, displaying soil moisture (y-axis) versus temperature anomaly (x-axis). Colored lines show the soil moisture-temperature values from each storyline. Scatter points represent analogue-derived values, with the darker points indicating analogue events that exceed the soil moisture and temperature thresholds of the corresponding storyline.

*The 2018 heatwave was very extreme, not only because of its circulation, but also because of its exceptionally preceding dry conditions (Rousi et al., 2023). In Figure 8, we explore the role of soil moisture in both approaches. In Fig. 8(a), we see how soil moisture is also affected by global warming, decreasing at a faster rate in future levels. The counterfactual storyline had a mean soil moisture of 62.9 kg/m2 during the main heatwave event, while in a +4K storyline, the soil moisture dropped to 54.9 kg/m2. In Figure 8(b), we show the bivariate distribution of soil moisture (SM) and temperature anomaly (T) associated with the flow analogues. The lines in this plot show the corresponding magnitudes in the storyline approach per global warming level. The bivariate comparison demonstrates that the analogues at each warming level occupy temperature(T)-soil moisture(SM) states similar to those in the corresponding storylines and that also their number of extreme heatwaves increases in their own warming levels when soil moisture is included as an additional condition in the analogue attribution (area of bivariate (SM - T) threshold exceedance shown as darker dots in Fig. 8(b)). Here, we can provide evidence that despite having similar atmospheric circulation patterns, the analogues do portray a larger spectrum of events that could emerge due to other underlying conditions, like enhanced soil moisture deficit. It is possible that due to the increasing thermodynamic drying of the future simulations, the role of the atmospheric circulation intensity becomes less important, as more of the analogues also have drought conditions.*

*Furthermore, the increase of extreme heatwaves in the upper tail of the temperature (Figure 6) and bivariate SM-T (Figure 8) distributions does not come only from the shift in the mean of the distributions towards warmer temperatures due to global warming, but there is also an increase the variance in future levels (see label Figure 6), which could be associated to a strengthened feedback with soil moisture in the analogue catalogue. If one could condition the temperature distributions on the soil moisture of a storyline (i.e. p(SM=60, T) ), the variance would be largely reduced. This would lead unfortunately also to a smaller sample size, too small to perform significant attribution.*

*In sum, both attribution methods are conditioned on the observed dynamics and were made as comparable as possible. The fact that the upper tail of the global warming levels in MPI-ESM increases more prominently does not imply that the approaches disagree. The storyline approach provides a representation of how this single event would change in response to anthropogenic thermodynamic forcing (SSTs and GHGs). The flow analogue approach includes a wider range of heatwaves developing under similar dynamics but different interacting processes, including more extreme heatwaves, probably resulting from a reduced soil moisture availability. Even though these results may be model-dependent, it is worth emphasizing that both approaches rely on the same atmospheric model. The storylines are simulated with ECHAM6, the atmospheric component of the MPI-ESM model, while the flow-analogues are extracted from scenario simulations of MPI-ESM. Including additional models for comparison in future work could provide more detailed insights on this matter. "*

**Lines 408 –415:**

*"Warmer storylines of the 2018 heatwave are not as rare at their corresponding level of global warming, as the 2018 heatwave was in the present. While the heatwave intensities increase linearly per degree of global warming, their frequency exponentially increases. Specifically, the observed 2018 heatwave had a 2.2 °C intensity with a return period of 1-in-277 years in the present time, intensifying to 6.6 °C with a return period of 1-in-26 years in a +4K world.* **The reason for the increased intensification in future climates is subject to further studies. So far, we assume that these differences could emerge from the conservative representation of global warming in the storyline approach, soil-feedback mechanisms enhanced in the MPI-ESM grand ensemble that leads to a larger variance with global warming, and other processes accounted in the coupled model"**

**Lines 459 – 462:**

*"The major outcome of this study is the combination of two conditional attribution methods, namely storylines and analogues. In this way we can provide information how a specific historic extreme event would have occurred without anthropogenic warming and how it*

*might behave for future climate states in combination with statistics like probabilities of occurrence for classes of similar extremes...”*

**Lines 467 – 473:**

*“In Section 4.3.1 we discuss the possible reasons for this increase in probability. According to Feser et al. (2024), the spectrally nudged storyline approach is conservative as it represents the influence of anthropogenic global warming solely through information stored in sea surface temperature and greenhouse gases, while other variables, which are likely also influenced by human activity -like aerosols- are not taken into account, representing climate change more cautiously. **This may restrict heatwave intensity, which accompanied by the role of soil moisture in temperature variance and other unknown processes coming from the coupled model may explain the increase in the upper tail distribution of the heatwaves detected in the analogue approach, compared to the simulated storylines.”***

**Lines 483-489:**

*“**Naturally, it is not only the atmospheric circulation pattern and global warming that define the characteristics of a heatwave.** For such reason, not all flow-analogues evolve into a heatwave, **and others get much more intense than the observed one.** Many other factors and interactions play a role, **like preceding drought conditions.** Moreover, by combining the storyline and flow-analogue approaches, we are not only able to project heatwaves under future warmer scenarios but also to assess their likelihood. **This synthesis allows us to state that future events with a similar blocking system to the 2018 heatwave might become more extreme and more frequent in future climates than the 2018 event was in the present**, highlighting a critical shift in environmental risks as global temperatures rise.”*

a)  I can imagine possible physical explanations for this phenomenon: 2018 stood out not just by the circulation during the HW but also by the preceding precipitation anomaly. This led to dryer soils during the HW which exacerbated the impacts of the 2018 circulation. Due to the increasing thermodynamic drying of the future simulations, the role of the circulation intensity becomes less important, as more and more of the analogues also have drought conditions. You could test this by comparing the temperature anomaly of the 2018 circulation to the other summer temperature anomalies within each storyline scenario. If there is such an underlying physical explanation, the temperature

anomalies of the 2018 circulation would move closer to the mean with an increasing GWL.

**Authors' Response:** Thanks for your interpretation of the possible underlying mechanisms behind the increased probability of the 2018 heatwave in warmer worlds. Both in our discussion with Reviewer 1 and in the manuscript, we have analyzed the role of soil moisture in both approaches. Your argument aligns with the finding in our new Fig. 8: analogues show soil conditions even drier than the corresponding storylines. Specifically, we like your interpretation, which has been included in the manuscript.

**Lines**                                                           **369-371:**
*"It is possible that due to the increasing thermodynamic drying of the future simulations, the role of the atmospheric circulation intensity becomes less important, as more of the analogues also have drought conditions."*

In the previous point, we addressed the discussion on this point as a possible reason for the increase in extreme heatwaves, together with Reviewer 1's input highlighting that soil moisture deficit leads to an increase in variance of the analogue's distributions.

**Lines 467 – 473:**

*"**In Section 4.3.1 we discuss the possible reasons for this increase in probability.** According to Feser et al. (2024), the spectrally nudged storyline approach is conservative as it represents the influence of anthropogenic global warming solely through information stored in sea surface temperature and greenhouse gases, while other variables, which are likely also influenced by human activity -like aerosols- are not taken into account, representing climate change more cautiously. **This may restrict heatwave intensity, which accompanied by the role of soil moisture in temperature variance and other unknown processes coming from the coupled model may explain the increase in the upper tail distribution of the heatwaves detected in the analogue approach, compared to the simulated storylines."***

**Lines 483-489:**

*"**Naturally, it is not only the atmospheric circulation pattern and global warming that define the characteristics of a heatwave.** For such reason, not all flow-analogues evolve into a heatwave, **and others get much more intense than the observed one.** Many other factors and interactions play a role, **like***

*preceding drought conditions. Moreover, by combining the storyline and flow-analogue approaches, we are not only able to project heatwaves under future warmer scenarios but also to assess their likelihood. **This synthesis allows us to state that future events with a similar blocking system to the 2018 heatwave might become more extreme and more frequent in future climates than the 2018 event was in the present,** highlighting a critical shift in environmental risks as global temperatures rise."*

b) However, I think there might be a different explanation for the observed pattern, as I don't agree with the statement that the "approaches are physically consistent, as they rely on the same model" (L332). While the models share the atmospheric component, they differ in the ocean (fully coupled, vs AMIP based on observations scaled to warming patterns from large ensemble), and more importantly in the atmospheric dynamics (ECHAM6 vs nudged-to-NCEP). Small differences in climatology or warming rate between storylines and fully coupled grand ensemble would translate to a shift in the slope of the stars in Fig. 7b. I think this is important to mention (or you could compare the ECHAM_SN climatology from (Schubert-Frisius et al., 2017) to the climatology in the grand ensemble?).

**Authors' Response:**

We agree with the reviewer regarding the descriptions of the differences between the model set-ups. We could unfortunately not couple the ocean, as, not being nudged due to a lack of high-quality observation data to nudge to, it would deteriorate and constantly work against the atmosphere which is nudged (and we did not want to have to frequently reinitialize the model because of that). The existing difference in atmospheric dynamics between storylines and the grand ensemble should not have much effect in this study as we compare only similar weather patterns. Both attribution methods are conditional on the observed dynamics and were made as comparable and similar as possible, that's what we meant with 'physically consistent'. We included in the text the use of climatology of Schubert-Frisius et al. (2017) in this study. We compared this climatology against the one of the grand ensemble (Fig. R1), showing similar magnitudes over the summer months used in this study (June-August). Finally, we think that the differences between climatologies should not be relevant, since we are comparing anomalies between equivalent datasets. For each ensemble member we calculate the temperature anomalies between future climate and the corresponding climatology of the same member.

[Figure]

**Fig. R1. Climatology comparison between storylines and MPI-ESM over Central Europe.** Here we show the 95th percentile of the climatology (1985-2014) used to define a heatwave. Thin light blue lines correspond to each ensemble member, and the dark blue lines correspond to the ensemble mean. The orange lines correspond to the 95th percentile of the same climatological period spectrally nudged to NCEP reanalysis data.

**Lines 138-140:**

*"We use the long-term ECHAM_SN simulation also spectrally nudged towards NCEP reanalysis data (Schubert-Frisius et al., 2017) as a climatological period (1985-2014) consistent with our simulations."*

2)  Selection of the analogs: From how I understand your approach, the main idea is to loosen the conditioning around a storyline in order to consider 'similar' events as well. However, it is not trivial to me to what extent the definition of similar will influence the outcome of this analysis. My impression from (Rousi et al., 2023) is, that the dynamical conditions that caused the 2018 HW were quite exceptional. It's understandable that these conditions must be loosened, but while your framework relies heavily on the concept of 'similar' circulation analogues, you set the definition in a way that defines the 2018 circulation as a 1-in-4 years event. Resulting from this, 83.5% of the analogues don't produce a HW under the factual climate. I'm not arguing that these conditions need to be stricter, but I would like to see a discussion on how the strictness in the conditioning of the analogues impacts a) the quantitative results, and b) how much the interpretation of the results hinges on the concept of 'similarity'.
    I also agree with point 2b by reviewer #1, and think that a conditioning on the temporal pattern would pull away some weight from finding analogues with the right

intensity (As seen in Fig. S4 the analogues are a bit on the weak side. For me, it would have helped to see some examples of the analogues and their related temperature patterns).

**Authors' Response:** We agree that the definition of "similar" atmospheric circulation is central to our framework. As also discussed in response to Reviewer 1 (point 2b), we designed the analogue detection algorithm following advice from our colleague (and flow-analogue method developer (Zorita and von Storch, 1999)) Eduardo Zorita. Our analogue-selection method intentionally balances two competing needs: retaining the temporal evolution of the 2018 pattern, and keeping a sufficiently large pool of good analogues. As also argued in our previous response to Reviewer 1, we can apply this 2-step filter due to the large sample we have available (92000 days: 92 summer days x 20 years x 50 members). Still, we did a brief analysis to check what if we looked for analogues based on their 5-day mean pattern instead of their sequence. In Fig. R2 we show that the temperature anomaly distributions are unaffected in shape, but using a 5-day mean provides a larger sample. It is very likely that the analogues identified by the 5-days sequence threshold are a subset of the ones identified by the 5-day mean threshold. In a way, all analogues identified through a sequence contain temporal evolution and high mean correlation, while the ones identified by the mean pattern contain high correlation and larger sample while including patterns evolving differently to the one of interest. Therefore, given that we obtain a large enough sample and more detailed information using our initial method, we find our more adequate for the aim of our study.

Evidently, quantitative results (ex: number of analogues) are sensitive to the thresholds and definitions, but we focus on the relative changes between global warming levels.

[Figure]

**Fig. R2 Analogue detection conditioning.** The blue distribution corresponds to the analogues identified in the MPI-ESM GE as described in the manuscript using a 5-days sequence as a reference pattern. The red distribution corresponds to the analogues identified using a 5-day mean of the reference pattern. N shows the number of analogues.

**3)** Forced circulation changes: The approach relies on a correct representation of P(D) by the grand ensemble. The fact that this is a topic of high uncertainty is one of the main reasons for using the storyline approach. While (Vautard et al., 2023) don't go into a separation of forced and unforced dynamical trends, their results suggest, that no member of the MPI-ESM GE is able to reproduce the trends in European circulation patterns related to summer heatwaves. Since the advantage of the proposed framework is its consideration of circulation changes, **I think it should be discussed what we can realistically expect from CMIP6 models (Shaw et al., 2024).**

**Authors' Response:**

We have included a brief discussion in the manuscript.

**Lines 439 – 446:**

*"Recent studies have shown that all nearly all CMIP6 model simulations fail to fully capture the accelerated trend in western European heat extremes, mostly by underestimating or totally missing the circulation-induced contribution related to more frequent southerly flow anomalies (Kornhuber et. al, 2024, Singh et al., 2023, Vautard et al. 2023). It has to be kept in mind that our combined attribution is designed to assess the impact of climate change on individual observed extreme events. While we account for possible dynamical changes in the future, we only do so by estimating the future probability of an event that is dynamically similar to the observed one. Our method therefore cannot provide more general inferences about expected changes in the dynamical contribution – either forced or from internal variability – that may dampen or amplify future heat extremes."*

To summarize: While there are currently limitations to the feasibility of the proposed approach, which are mostly based on the physical consistency between the two elements, I don't think that these constraints devalue the usability of this framework.

**Authors' Response:**

Thank you for the positive evaluation of our framework. We now tried to better explain and discuss the limitations of the feasibility of the method.

Minor comments:

1) It looks like the model output and ERA5 have a 12h offset (while they both refer to the daily means). You could correct this to make the plot look nicer.

**Authors' Response:**

Thanks for your observation, we've corrected Figure 2.

[Figure]

**Figure 2. The 2018 Central European heatwave...**

2) The reference by (van Garderen et al., 2021) has the wrong year.
   **Authors' Response:**
   Thank you, we have corrected the reference.

3) The event definition (blue box in Fig. 2) is based on the impacts of the 2018 circulation, but the key point is to loosen the circulation restrictions, thereby including circulation analogues with different spatial temperature patterns. So I wonder if it would make sense to use a bigger box for the second part of the analysis?
   **Authors' Response:**
   We consider that keeping the same region of study is crucial for the consistency between methods, otherwise we would not be attributing changes in the same region. Besides, in the same box (Central Europe), different spatial temperature patterns can emerge. This is evidenced by the obtained temperature distributions, where many different magnitudes are observed, coming from a wide range of possible spatial patterns occurring under similar dynamics.

Best regards,

Istvan Dunkl

References:

van Garderen, L., Feser, F., and Shepherd, T. G. (2021). A methodology for attributing the role of climate change in extreme events: a global spectrally nudged storyline, Natural Hazards and Earth System Sciences, 21, 171–186, https://doi.org/10.5194/nhess-21-171-2021.

Otto, F. E. (2017). Attribution of weather and climate events. *Annual Review of Environment and Resources*, *42*, 627–646. https://doi.org/10.1146/annurev-environ-112621-083538

Schubert-Frisius, M., Feser, F., Storch, H. von, and Rast, S (2017). Optimal Spectral Nudging for Global Dynamic Downscaling, Monthly Weather Review, 145, 909–927, https://doi.org/10.1175/MWR-D-16-0036.1.

Shaw, T. A., Arblaster, J. M., Birner, T., Butler, A. H., Domeisen, D. I. V., Garfinkel, C. I., Garny, H., Grise, K. M., and Karpechko, A. Yu. (2024). Emerging Climate Change Signals in Atmospheric Circulation, AGU Advances, 5, e2024AV001297, https://doi.org/10.1029/2024AV001297.

Vautard, R., Cattiaux, J., Happé, T., Singh, J., Bonnet, R., Cassou, C., Coumou, D., D'Andrea, F., Faranda, D., Fischer, E., Ribes, A., Sippel, S., and Yiou, P (2023).Heat extremes in Western Europe increasing faster than simulated due to atmospheric circulation trends, Nat Commun, 14, 6803, https://doi.org/10.1038/s41467-023-42143-3.

Rousi, E., Fink, A. H., Andersen, L. S., Becker, F. N., Beobide-Arsuaga, G., Breil, M., Cozzi, G., Heinke, J., Jach, L., Niermann, D., et al. (2023). The extremely hot and dry 2018 summer in central and northern Europe from a multi-faceted weather and climate perspective, *Natural Hazards and Earth System Sciences, 23,* 1699–1718. https://doi.org/10.5194/nhess-23-1699-2023

Zorita, E., & von Storch, H. (1999). The Analog Method as a Simple Statistical Downscaling Technique: Comparison with More Complicated Methods. *Journal of Climate,* *12(8),* 2474-2489. https://doi.org/10.1175/1520-0442(1999)012<2474:TAMAAS>2.0.CO;2.

Kornhuber, K., Bartusek, S., Seager, R., Schellnhuber, H. J., & Ting, M. (2024). Global emergence of regional heatwave hotspots outpaces climate model simulations. Proceedings of the National Academy of Sciences, 121(49), e2411258121. https://doi.org/10.1073/pnas.2411258121

Singh, J., Sippel, S., & Fischer, E. M. (2023). Circulation dampened heat extremes intensification over the Midwest USA and amplified over Western Europe. Communications Earth & Environment, 4(1), 432. https://doi.org/10.1038/s43247-023-01096-7